# The ion channel function of polycystin-1 in the polycystin-1/polycystin-2 complex

Zhifei Wang[1], Courtney Ng[1], Xiong Liu[2] (ID), Yan Wang[1], Bin Li[1], Parul Kashyap[1], Haroon A Chaudhry[1], Alexis Castro[1], Enessa M Kalontar[1], Leah Ilyayev[1], Rebecca Walker[3], R Todd Alexander[4], Feng Qian[3], Xing-Zhen Chen[2] & Yong Yu[1,*] (ID)

## Abstract

Autosomal dominant polycystic kidney disease (ADPKD) is caused by mutations in *PKD*1 or *PKD*2 gene, encoding the polycystic kidney disease protein polycystin-1 and the transient receptor potential channel polycystin-2 (also known as TRPP2), respectively. Polycystin-1 and polycystin-2 form a receptor–ion channel complex located in primary cilia. The function of this complex, especially the role of polycystin-1, is largely unknown due to the lack of a reliable functional assay. In this study, we dissect the role of polycystin-1 by directly recording currents mediated by a gain-of-function (GOF) polycystin-1/polycystin-2 channel. Our data show that this channel has distinct properties from that of the homomeric polycystin-2 channel. The polycystin-1 subunit directly contributes to the channel pore, and its eleven transmembrane domains are sufficient for its channel function. We also show that the cleavage of polycystin-1 at the N-terminal G protein-coupled receptor proteolysis site is not required for the activity of the GOF polycystin-1/polycystin-2 channel. These results demonstrate the ion channel function of polycystin-1 in the polycystin-1/polycystin-2 complex, enriching our understanding of this channel and its role in ADPKD.

**Keywords** autosomal dominant polycystic kidney disease; gain-of-function; ion channel; polycystin-1; polycystin-2
**Subject Category** Membranes & Trafficking

## Introduction

Polycystic kidney disease (PKD) proteins and transient receptor potential (TRP) polycystin (TRPP) proteins form receptor/ion channel protein complexes, playing diverse roles in sensory systems [1,2]. The PKD protein polycystin-1 (PC1) and TRPP protein polycystin-2 (PC2) form a complex, which is localized on primary cilia of renal epithelial cells and is essential for renal tubular differentiation [2]. Mutations in the gene of either protein, *PKD1* or *PKD2*, cause autosomal dominant polycystic kidney disease (ADPKD), one of the most common inherited human diseases which affect one in every 400–1,000 individuals [2–5]. ADPKD is characterized by the formation of fluid-filled renal cysts, with more than half of patients progressing to renal failure. Mutations in *PKD1* account for ~80% of clinically identified cases, which are usually more severe than those caused by mutations in *PKD2* [6]. How these mutations lead to ADPKD is unclear, and the molecular mechanism of the function of the PC1/PC2 complex is largely unknown. It is generally believed that in the PC1/PC2 complex, PC1 functions as a receptor to sense unknown extracellular stimuli, such as mechanical force or chemical ligands, and couples it with intracellular signaling through $Ca^{2+}$ influx conducted by the PC2 channel [1,2,7,8]. However, thus far the *in vivo* stimuli of this complex remain unknown, although previous studies suggested fluid flow as the potential stimulus [8] and Wnt proteins as candidate ligands [9]. The $Ca^{2+}$ conductance by PC2 or PC1/PC2 complex is also hotly debated [7,9–16].

Currently, we know much more about the function of the PC2 protein than PC1. PC2 has six transmembrane domains (S1-S6) and intracellular N- and C-termini, similar to other TRP cation channels (Fig 1A) [17,18]. PC2 forms homotetramers in the absence of PC1 [19–21], which function as non-selective cation channels [13–15,22,23]. In contrast, despite intensive studies for over 20 years, the roles of PC1 are largely unknown. PC1 has eleven transmembrane domains, a short intracellular C-terminal tail, and a large extracellular N-terminus (Fig 1A) [24,25]. The N-terminus of PC1 contains well-recognized motifs involved in protein–protein, protein–saccharide, and protein–ligand interactions. Thus, PC1 is generally thought to function as a cell surface receptor [2,3,26,27]. Indeed, PC1 has significant similarities to the adhesion G protein-coupled receptors (aGPCRs), a large group of proteins involved in many signaling pathways [28]. Like aGPCRs, PC1 contains a GPCR autoproteolysis-inducing (GAIN) domain located upstream of the

1   Department of Biological Sciences, St. John's University, Queens, NY, USA
2   Department of Physiology, Membrane Protein Disease Research Group, Faculty of Medicine and Dentistry, University of Alberta, Edmonton, AB, Canada
3   Division of Nephrology, Department of Medicine, University of Maryland School of Medicine, Baltimore, MD, USA
4   Departments of Pediatrics and Physiology, University of Alberta, Edmonton, AB, Canada
    *Corresponding author. Tel: +1 718 990 1654; E-mail: yuy2@stjohns.edu

first transmembrane domain and undergoes autoproteolytic cleavage at a G protein-coupled receptor proteolysis site (GPS) within the GAIN domain [29,30]. GPS cleavage splits the protein into an extracellular N-terminal fragment (NTF) and a C-terminal fragment (CTF); the latter contains the transmembrane domains and the intracellular C-terminal tail [30]. The cleaved NTF and CTF tether together through a non-covalent interaction [30,31], and GPS cleavage is essential for *in vivo* function of PC1 in mice [29,32]. Consistently, studies demonstrate that PC1 has a C-terminal G protein binding site [33,34] and the function of PC1 is linked to G protein signaling [35–38]. Thus, PC1 may function as an atypical GPCR, although we do not know if this function is PC2-dependent.

There are several lines of evidence suggesting that PC1 functions as an ion channel subunit in the PC1/PC2 complex. First, the last six transmembrane domains (S6-S11) of PC1 share sequence homology with TRPP channels (Fig 1A) [24,25]. Second, our biochemical and biophysical studies found that the PC1/PC2 complex contains one PC1 and three PC2 subunits [39,40]. Thus, in this complex, PC1 may take the position of the fourth PC2 subunit and form the channel pore with the other three PC2 subunits. Third, our previous study showed that another PKD/TRPP complex, PKD1L3/TRPP3, shares the same 1:3 subunit stoichiometry and that the single PKD1L3 subunit functions as a channel-forming component [41]. PC1 may have a similar channel role in its complex with PC2. Fourth, the recently published cryo-EM structure of a transmembrane fraction of the human PC1/PC2 complex confirmed that this complex indeed has 1 (PC1): 3 (PC2) stoichiometry [42]. More importantly, it shows that the last six transmembrane domains of

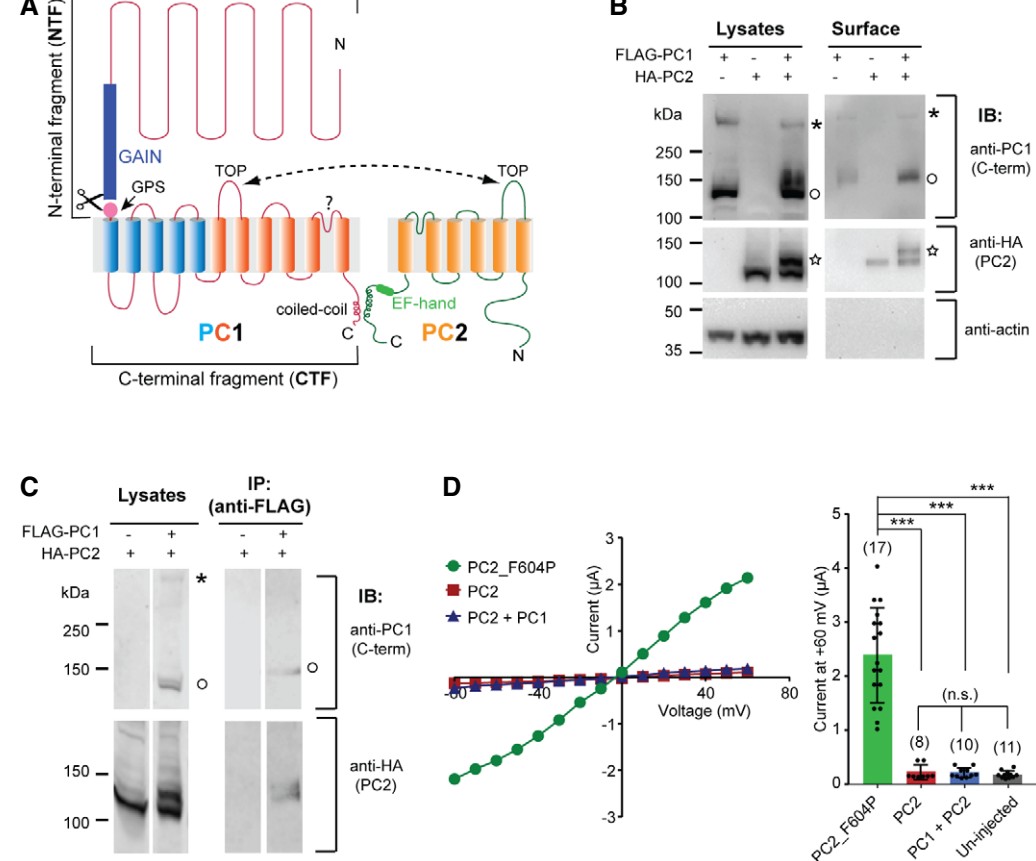

**Figure 1. PC1 and PC2 express in *Xenopus* oocytes but yield no channel current.**

A Transmembrane topology of PC1 and PC2 proteins. The two proteins associate at the C-terminus through the coiled-coil domains and the extracellular side via the TOP domains. The GAIN domain and the GPS site in PC1 and the EF-hand motif in PC2 are indicated. The last six transmembrane domains of PC1 (shown in orange) share sequence similarity with PC2.

B Western blot of oocyte lysate (left) and biotinylation-purified surface (right) samples showing the expression of PC1 and PC2 in *Xenopus* oocytes and enhanced surface trafficking of the PC1/PC2 complex compared to either protein expressed individually. Anti-PC1 C-terminus antibody [29] recognized both full-length (asterisk) and GPS-cleaved CTF (open circle) of PC1. A higher-glycosylated 130 kDa PC2 (star) band was only seen when PC1 is coexpressed.

C Co-IP followed by Western blot showing the association between PC1 and PC2 that were expressed in *Xenopus* oocytes. IP was done with an anti-FLAG antibody. Bands of full-length (asterisk) and GPS-cleaved CTF (open circle) of PC1 are indicated. Both 120 and 130 kDa bands of PC2 were seen in the IPed product.

D Representative current–voltage relationship (I–V) curves (left) and a scatter plot and bar graph (right) showing coexpression of WT PC1 and PC2 produced no current in TEVC recording. The current of the GOF PC2_F604P is included as a control. Currents at +60 mV are shown in the bar graph. Each point represents the recorded current from one oocyte. Oocyte numbers for scatter plot and bar graph are indicated in parentheses. Data are presented as mean ± SD (n.s.: not significant, ***$P < 0.001$, Student's *t*-test).

Source data are available online for this figure.

the single PC1 subunit assemble with three PC2 subunits into a TRP channel-like complex. This complex shares overall structural similarity to the homotetrameric PC2 channel, with the last two transmembrane domains (S10–S11) of PC1 directly participating in forming the presumed channel pore [42].

Despite evidence indicating that PC1 functions as an ion channel subunit in the PC1/PC2 complex, a solid conclusion could not be made since functional evidence is lacking. This is mainly due to the lack of knowledge on the channel activation mechanism, making functional analysis very challenging. To resolve this, we recently developed gain-of-function (GOF) mutants of the PC2 channel, including the reported F604P mutant [22]. PC2_F640P gave rise to robust whole-oocyte currents when expressed in *Xenopus laevis* oocytes [11,22,43]. The cryo-EM structure showed that the F604P mutation leads to twisting and rotation of the distal S6 helix and opening of the lower pore gate [43]. Interestingly, the structure of PC2_F604P is very similar to an open structure of TRPP3 [44], suggesting that the F604P mutation results in a conformational change reflecting the natural gating of PC2. In the current study, taking advantage of another GOF PC2 mutant, we were able to reliably record channel activity of PC1/PC2, study its ion permeability, and, for the first time, dissect the function of PC1 in this channel. We further explored the core channel unit in PC1 and how GPS cleavage affects PC1/PC2 channel activity. This study sheds light on the molecular mechanism of the PC1/PC2 function and ADPKD pathogenesis.

# Results

## Oocyte-expressed PC1 and PC2 retain their native characteristics

To investigate the function of PC1/PC2, we coexpressed PC1 and PC2 proteins in *Xenopus laevis* oocytes and evaluated their expression. Both proteins expressed very well as shown by Western blot (Fig 1B, left images), and two important features were noted. First, oocyte-expressed PC1 was cleaved at the N-terminal GPS site, recapitulating *in vivo* and mammalian cell expression results [29,30]. When detected with an antibody recognizing the PC1 C-terminus, a full-length band (above the 250 kDa marker) and a cleaved CTF band (~130 kDa) of PC1 were detected (Fig 1B, left top image), matching the pattern of PC1 cleaved at the GPS site [30]. As observed *in vivo* [29], the majority of oocyte-expressed PC1 was cleaved, since a greater amount of cleaved CTF than full-length protein was detected (Fig 1B). Second, an additional PC2 band was identified at 130 kDa when PC1 was coexpressed, contrasting the single band at ~120 kDa when PC1 was absent (Fig 1B, left middle image). It has been previously reported that the 130-kDa PC2 band is a higher-glycosylated (EndoH-resistant) form of PC2 which stays in complex with PC1 in cilia of native tissues, while the 120 kDa PC2 only contains EndoH-sensitive glycosylation [45]. Thus, both PC1 and PC2 retain their native features after being expressed in *Xenopus* oocytes, validating the feasibility of studying the function of the PC1/PC2 complex in this system. The cell surface biotinylation experiment reveals that the PC1/PC2 complex traffics to the plasma membrane in oocytes (Fig 1B). In co-immunoprecipitation (co-IP) experiments, the association between PC1 and both forms of PC2 was detected (Fig 1C). Proteins in our surface biotinylation or co-IP samples sometimes ran at a higher molecular weight than the

same proteins in lysate samples (Fig 1B and C), which is most likely due to the high concentration of salt used in elution.

Although the PC1/PC2 complex trafficked to the cell surface in oocytes, coexpression of the two proteins did not lead to a measurable current in recording with two-electrode voltage clamp (TEVC), while the previously described GOF PC2 mutant, PC2_F604P [22,43], gave rise to robust currents in oocytes (Fig 1D). This result suggests that the wild-type (WT) complex channel stays in a closed state within the oocyte plasma membrane.

We hypothesized that coexpression of PC1 with PC2_F604P might lead to an open PC1/PC2_F604P complex channel. However, the coexpression of PC1 greatly reduced the current (Fig EV1). The current reduction may be due to lower activity of the complex channel formed by PC1 and PC2_F604P. A similar dominant negative effect was previously seen when coexpressing WT PC2 with PC2_F604P [22], which is likely due to the WT subunits in the pore obstructing the F604P-induced gating process. The effect of PC1 on complex channel activity suggests that the association of the PC1 subunit also changes the channel pore of PC2_F604P. Since the much lower channel activity of PC1/PC2_F604P renders it unsuitable for studying channel function, we generated new GOF PC2 mutants intending to find one that leads to the opening of the PC1/PC2 channel.

## Mutation at the lower gate generates a new GOF PC2 channel

To generate new GOF mutants, we focused on the amino acids forming the lower gate of PC2. Structural studies by cryo-EM [19–21] and recent functional study [43] show that L677 and N681 are either physical or functional lower gate residues (Fig 2A). Mutating both amino acids to alanine (L677A/N681A, named "AA") produced significant currents in a bath solution containing 100 mM $Na^+$ and 2 mM $Ca^{2+}$ when the mutant channel was expressed in oocytes (Fig 2B). The current size of PC2_AA is roughly double the size of that of PC2_F604P. Just like that of PC2_F604P [22], the current of PC2_AA is outwardly rectifying. The inward current, presumably carried by $Na^+$, is inhibited when 2 mM extracellular $Ca^{2+}$ is present (compare the I–V curves in Fig 2B and C). This suggests that at negative voltages, but to much less extents at depolarized voltages, extracellular $Ca^{2+}$ is attracted to the pore, which blocks $Na^+$ entry, a common phenomenon in cation channels. Because the majority of the intracellular cations in oocytes are $K^+$ [46], the outward current mediated by PC2_AA channel should mainly be due to $K^+$ efflux.

We further tested the channel's $Ca^{2+}$ permeability by replacing all cations in the bath solution with 70 mM $Ca^{2+}$. In this solution, the PC2_F604P current was almost completely inhibited, indicating that it has no $Ca^{2+}$ permeability or that it is too low to be detected with this method. However, PC2_AA gave rise to robust currents in this solution (Fig 2D). The current–voltage relationship (I–V) curve of this current has an unusual trough at strong hyperpolarization, due to slow development of the currents upon application of the first strong negative voltage (Appendix Fig S1). We postulate that the currents of PC2_AA in 70 mM $Ca^{2+}$ are combined PC2 current with $Cl^-$ currents conducted by endogenous calcium-activated chloride channels (CaCC) in oocytes [47–51]. CaCC can be activated by $Ca^{2+}$ influx and has been used as an amplifying system to monitor $Ca^{2+}$ influx such as store-operated $Ca^{2+}$ entry (SOCE) [52]. Indeed,

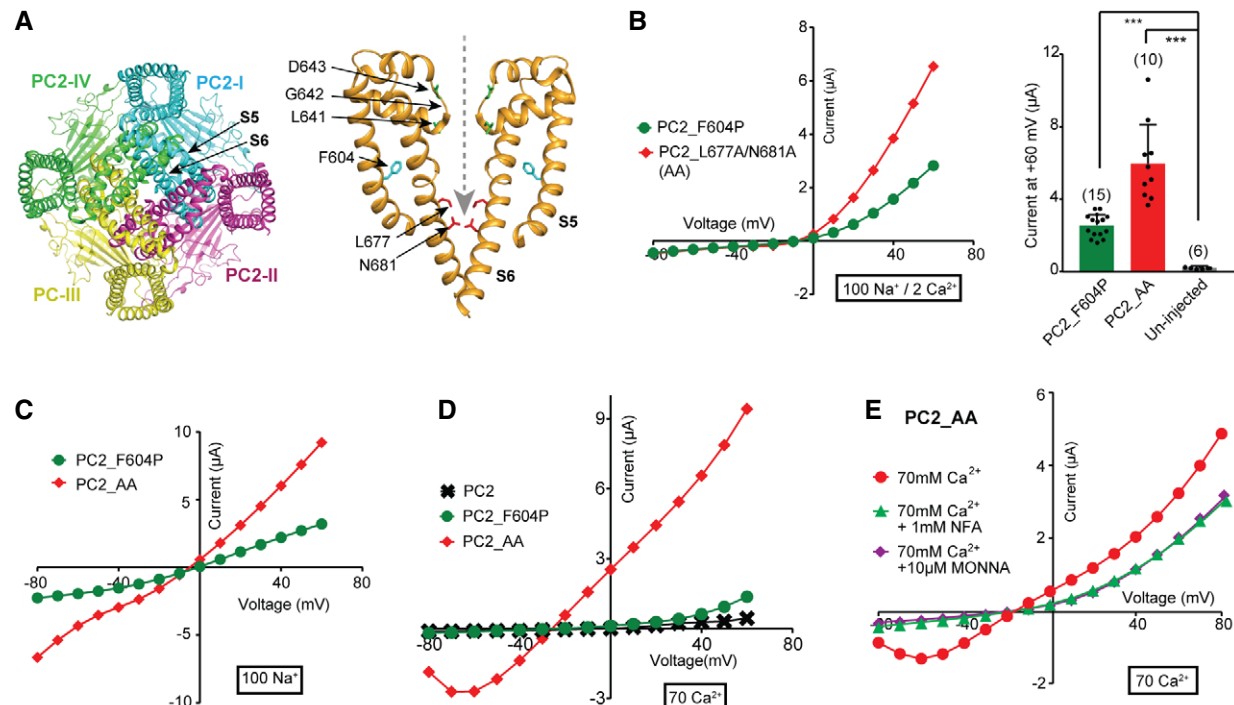

**Figure 2. PC2_L677A/N681A (AA) mutant is a new GOF channel.**

A   Cryo-EM structure of PC2. Left: Bottom view of PC2 homomeric tetramer showing the pore-lining S5-S6 from each subunit assembles into the channel pore. Right: Side view of S5-S6 from two subunits showing the L677 and N681 (in red) contributes to the restriction at the lower gate. F604P (in cyan) on S5 and the three residues (in green) forming the selectivity filter are also indicated. The structure shown here is previously reported with Protein Data Bank (PDB) code 5T4D [20]. A gray dashed line indicates the path of the ion flow.

B   Representative I–V curves (left) and a scatter plot and bar graph (right) showing that the PC2_AA is a GOF mutant and gave rise to a larger current than PC2_F604P. Scatter plot and bar graph show the average current sizes at +60 mV. The cations included in the bath solution, 100 mM $Na^+$ and 2 mM $Ca^{2+}$ in this case, are indicated by the thick-lined boxes here and in all the following figures. Oocyte numbers for bar graph are indicated in parentheses. Data are presented as mean ± SD in bar graph (\*\*\*$P < 0.001$, Student's *t*-test).

C   Representative currents of PC2_F604P and PC2_AA mutants in the divalent ion-free bath solution, which contains 100 mM $Na^+$.

D   Representative currents of indicated WT and mutants of PC2 in a bath solution containing 70 mM $Ca^{2+}$.

E   Calcium-activated chloride channel (CaCC) blocker MONNA (10 μM) or niflumic acid (NFA) (1 mM) partially blocked the current recorded from the PC2_AA-injected oocytes in the 70 mM $Ca^{2+}$ bath solution.

Source data are available online for this figure.

including chloride channel blockers, niflumic acid (NFA) or N-((4-methoxy)-2-naphthyl)-5-nitroanthranilic acid (MONNA) [53,54], in the bath solution, dramatically reduced the current of PC2_AA in 70 mM $Ca^{2+}$ (Fig 2E). These results confirm that a large portion of the currents in 70 mM $Ca^{2+}$ solution is conducted by CaCC. Since the CaCC current was not seen in WT and PC2_F604P-injected oocytes (Fig 2D), the elevated CaCC activity in PC2_AA-injected oocytes is not produced by higher expression of the CaCC channel stimulated by PC2 expression. Thus, the results imply that PC2_AA is $Ca^{2+}$-permeable and that $Ca^{2+}$ influx through these channels led to CaCC activation. Our data suggest that the CaCC current can be used as a readout for measuring the relatively small $Ca^{2+}$ conductance of PC2 in *Xenopus* oocytes, which we have applied in the following experiments.

## PC1/PC2_AA is a GOF PC1/PC2 complex channel

We then coexpressed PC2_AA with full-length PC1 in *Xenopus* oocytes to determine whether they will form a GOF complex channel.

The injected cRNA molar ratio of PC1 to PC2_AA was 1.5:1 to minimize the homomeric PC2_AA channel formation. Considering the 1:3 ratio of PC1 to PC2 in PC1/PC2 complex, the PC1 RNA we injected is 4.5-fold oversaturated. A later experiment confirmed that in this experimental condition, all detected PC2 are in the complex with PC1 (shown in Fig 7D). In divalent ion-free solution, oocytes expressing both PC1 and PC2_AA gave rise to similar I–V curves, in terms of both the voltage dependence and current magnitude, to those expressing PC2_AA alone (Fig 3A). The only difference we noticed is that the reversal potential (RP) was slightly shifted toward a positive voltage. However, when 2 mM $Ca^{2+}$ was included in the bath solution, compared to the outwardly rectifying current of PC2_AA, the current of PC1/PC2_AA shows almost no rectification, indicating that the PC1/PC2_AA complex has much less or no extracellular $Ca^{2+}$ blocking (Fig 3B). The inward current of PC1/PC2_AA is carried by $Na^+$ influx and has no detectable CaCC component since adding MONNA led to no change of the current (Fig 3C). Similar to $Ca^{2+}$, 2 mM $Mg^{2+}$ also abolished the inward current of the PC2_AA channel but not of the PC1/PC2_AA channel (Appendix Fig S2).

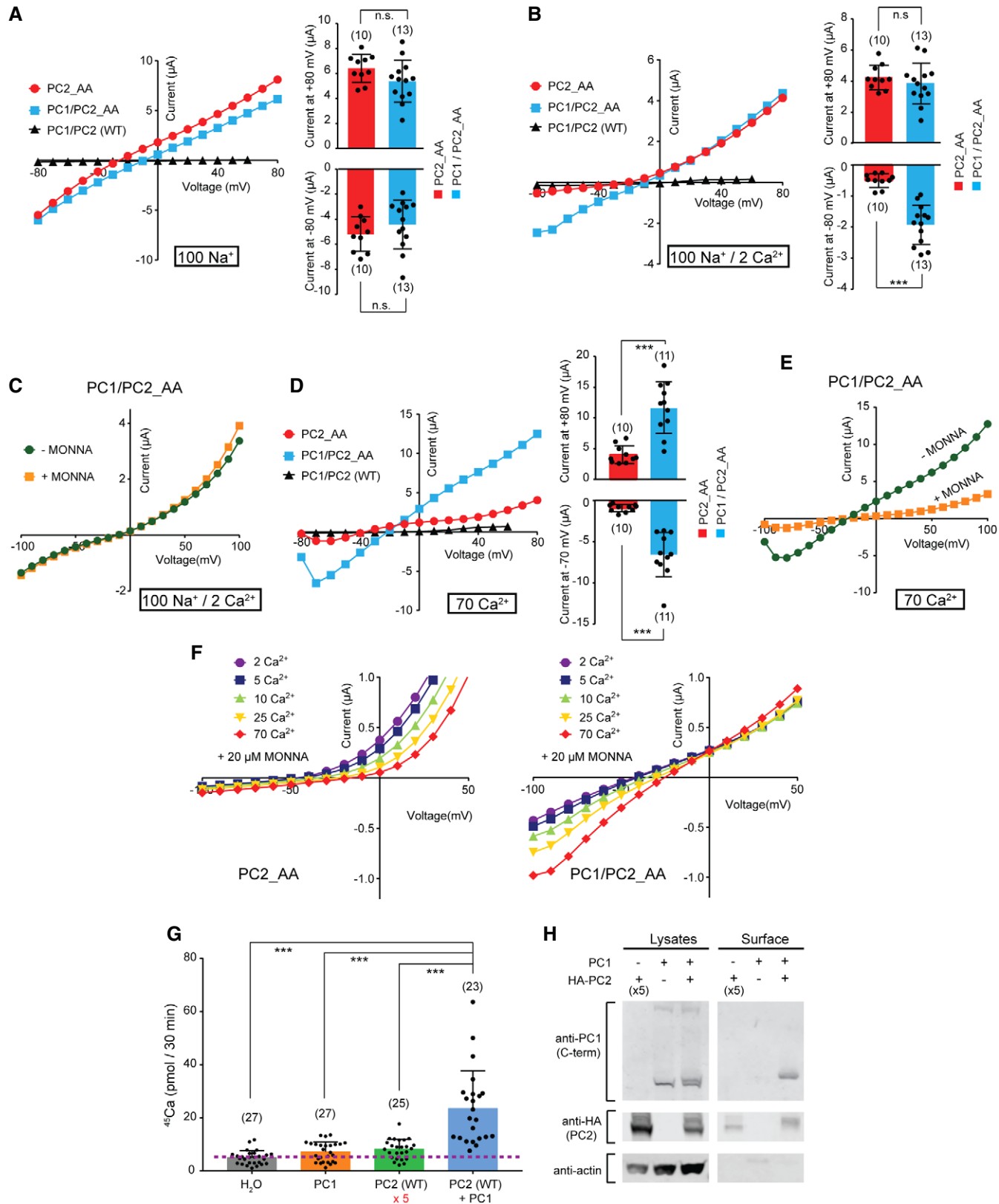

**Figure 3.**

**Figure 3. The PC1/PC2_AA complex channel has greater Ca$^{2+}$ permeability than the homomeric PC2_AA channel.**

A, B    Representative I–V curves (left) and scatter plot and bar graphs (right) showing the currents from oocytes expressing PC2_AA alone, PC1 with PC2_AA, and PC1 with WT PC2, in bath solutions containing 100 mM Na$^+$ (A) and 100 mM Na$^+$ and 2 mM Ca$^{2+}$ (B). Scatter plot and bar graphs show currents at +80 mV and −80 mV. Oocyte numbers are indicated in parentheses. Data are presented as mean ± SD in bar graph (n.s.: not significant, ***$P < 0.001$, Student's *t*-test).

C    Representative I–V curves of PC1/PC2_AA recorded in a bath solution containing 100 mM Na$^+$ and 2 mM Ca$^{2+}$ solution in the absence or presence of 10 μM MONNA.

D    Representative I–V curves (left) and scatter plot and bar graphs (right) showing the currents from oocytes expressing PC2_AA alone, PC1 with PC2_AA, and PC1 with WT PC2, in bath solutions containing 70 mM Ca$^{2+}$. Scatter plot and bar graphs show currents at +80 mV and −70 mV. Oocyte numbers are indicated in parentheses. Data are presented as mean ± SD in bar graph (***$P < 0.001$, Student's *t*-test).

E    Representative I–V curves of PC1/PC2_AA recorded in a bath solution containing 70 mM Ca$^{2+}$ solution in the absence or presence of 10 μM MONNA.

F    Representative currents of PC2_AA (left) and PC1/PC2_AA (right) in solutions containing the indicated Ca$^{2+}$ concentrations. All solutions contain 20 μM of MONNA to block CaCC current. Corresponding concentrations of NMDG$^+$ were added to compensate the osmolarity. See Fig EV2 for currents of the full voltage scale and the details of solutions used.

G    Scatter plot and bar graph showing the higher radiolabeled $^{45}$Ca uptake rate of oocytes expressing the PC1/PC2_AA complex channel compared to that of oocytes expressing PC1 alone or PC2 alone. For oocytes injected with PC2 alone, five times more concentrated PC2 RNA was injected to increase its surface expression. The purple dashed line indicates the background $^{45}$Ca uptake set by the measurement with the water-injected oocytes. Data were averaged from three independent experiments. Oocyte numbers are indicated in parentheses. Data are presented as mean ± SD in bar graph (***$P < 0.001$, Student's *t*-test).

H    Surface biotinylation followed by Western blot showing the expression levels of the indicated proteins in lysate and plasma membrane of oocytes in $^{45}$Ca uptake experiments.

Source data are available online for this figure.

In 70 mM Ca$^{2+}$ solution, the PC1/PC2_AA-expressing oocytes gave more than three times larger currents than that of the PC2_AA-expressing oocytes, although the same amount of PC2_AA RNA was injected in both groups (Fig 3D). Similar to the PC2_AA current, applying MONNA partially and significantly blocked the current of PC1/PC2_AA in 70 mM Ca$^{2+}$ solution, indicating the current was conducted by both PC1/PC2_AA and CaCC (Fig 3E). The latter was activated by Ca$^{2+}$ entry through the PC1/PC2_AA channel. Compared to PC2_AA, PC1/PC2_AA produced a similar-sized current in Na$^+$ solution (Fig 3A), but much larger current in 70 mM Ca$^{2+}$ solution (Fig 3D), indicating that PC1/PC2_AA channel has greater Ca$^{2+}$: Na$^+$ permeability ratio ($P_{ca}/P_{Na}$) than PC2_AA channel. To further prove it, we compared the inward currents of PC2 and PC1/PC2 in solutions containing varying Ca$^{2+}$ concentrations, in the presence of 20 μM of MONNA. No Na$^+$ was included in these solutions, and the osmolarity was compensated by adding NMDG$^+$, which is not permeable through these channels. In this condition, most of the inward currents should be carried by Ca$^{2+}$ influx, although we could not rule out the presence of residual chloride current since MONNA may not completely inhibit all CaCC. The results show that decent amount of inward currents can be recorded from the PC1/PC2_AA channel and the reversal potential shifted to the right when Ca$^{2+}$ concentration increased in the solution, indicating Ca$^{2+}$ permeability (Figs 3F and EV2). In contrast, almost no inward current was recorded from PC2_AA in all solutions, indicating a much smaller Ca$^{2+}$ permeability of this channel (Figs 3F and EV2). Also, we noticed that increasing extracellular Ca$^{2+}$ concentration also led to more blocking of outward current, which is carried mainly by potassium efflux, of PC2_AA, but not PC1/PC2_AA (Figs 3F and EV2). This is consistent with the fact that extracellular Ca$^{2+}$ blocks more of the PC2_AA channel. Although we have observed the difference on Ca$^{2+}$ influx between the two channels, we were not able to calculate the relative Ca$^{2+}$ permeability since the apparent reversal potentials are largely affected by leak currents when the channel's inward currents are very small, especially in the case of PC2_AA.

Since these results were obtained from using a GOF PC1/PC2 channel, we further confirmed the Ca$^{2+}$ permeability of the WT PC1/PC2 channel by $^{45}$Ca radiotracer uptake experiments [55] in oocytes expressing PC1 alone, PC2 alone, or both PC1 and PC2.

Due to the high sensitivity of this method, we were able to trace the small Ca$^{2+}$ influx through the WT channels, even with their low open probability. Five times more PC2 cRNA was injected for the PC2 alone condition to boost the surface expression of PC2 homomeric channel. The results showed that the $^{45}$Ca uptake rate of the PC1- or PC2-injected oocytes was only slightly higher than H$_2$O-injected oocytes, while that of oocytes expressing PC1/PC2 was 6.72 ± 3.25-fold of that of the PC2-alone-expressing oocytes after background subtraction (Fig 3G). Under the designed experimental conditions, the total amount of surface-expressed PC2 in the PC1/PC2 sample is about 1.5-fold of that in the PC2 alone sample (Fig 3H, measured with ImageJ). Therefore, our data from this experiment strongly support that the WT PC1/PC2 channel has higher Ca$^{2+}$ permeability than the PC2 channel. We noticed that expressing the channel proteins usually depolarizes the membrane potential of oocytes to higher than −25 mV. Since the driving force for cation influx will be greater under a more negative membrane potential, the Ca$^{2+}$ permeability difference between PC1/PC2 and PC2 channels under physiological conditions would be greater than what we observed in the $^{45}$Ca uptake experiment.

## Both the CTF and the last six transmembrane domains are sufficient for ion channel function of PC1 in the PC1/PC2 complex

Cleavage at the GPS site dissects PC1 into an NTF (G24-L3040 in mouse protein) and a CTF (T3041-T4293). The latter contains all 11 transmembrane domains and the intracellular C-terminal tail (Fig 4A) [30]. A previous study reported that a significant amount of NTF detaches from the CTF *in vivo*, indicating that each isolated fragment may function independently in some situation [56]. We tested whether PC1-CTF itself is sufficient for forming a channel complex with PC2 by coexpressing the PC1-CTF with PC2_AA in *Xenopus* oocytes. An Ig k-chain leader sequence was fused to the N-terminus of PC1-CTF to prompt its trafficking to the plasma membrane. In the solution containing 100 mM Na$^+$, PC1-CTF/PC2_AA-injected oocytes yielded almost the same current as that of the full-length PC1/PC2_AA channel (Fig 4B). Likewise, extracellular 2 mM Ca$^{2+}$ did not block the inward current of PC1-CTF/PC2_AA, similar to that of the full-length

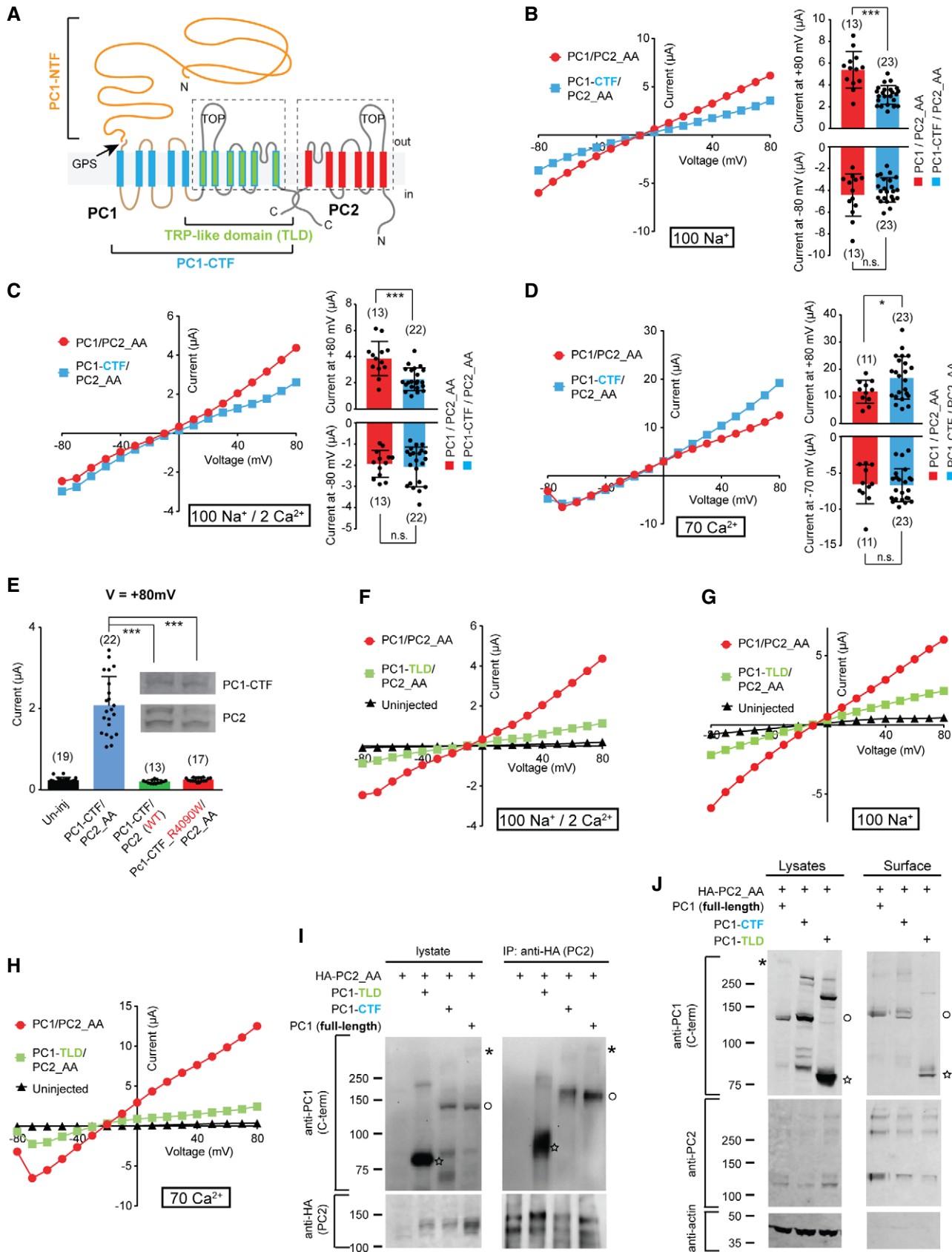

Figure 4.

◀

**Figure 4. Both the CTF and the TRP-like domain (TLD) are sufficient for ion channel function of PC1 in the PC1/PC2 complex.**

A   Transmembrane topology of PC1 and PC2 proteins, showing the NTF, CTF, and the TLD of PC1. The six transmembrane domains in TLD (indicated with the dashed line box) share sequence homology with the transmembrane domains of PC2.

B–D  Representative I–V curves (left) and scatter plot and bar graphs (right) showing the comparison between the currents of full-length PC1/PC2_AA with that of PC1-CTF/PC2_AA in bath solutions containing 100 mM $Na^+$ (B), 100 mM $Na^+$ and 2 mM $Ca^{2+}$ (C), or 70 mM $Ca^{2+}$ (D). Currents at both +80 mV and −80 (or −70) mV are displayed in the scatter plot and bar graphs. Oocyte numbers are indicated in parenthesis. Data are presented as mean ± SD in bar graph (n.s.: not significant, *$P < 0.05$; ***$P < 0.001$, Student's $t$-test).

E   Scatter plot and bar graph showing that the current of PC1-CTF/PC2_AA is completely abolished when WT PC2 was used, or after introducing R4090W mutation in the putative pore region of PC1-CTF. Inserted are Western blot images showing the expression of the corresponding proteins. Top: anti-PC1. Bottom: anti-PC2. Oocyte numbers are indicated in parenthesis. Data are presented as mean ± SD in bar graph (***$P < 0.001$, Student's $t$-test).

F–H  Representative I–V curves of oocytes injected with the indicated RNAs in a bath solution containing 100 mM $Na^+$ and 2 mM $Ca^{2+}$ (F), 100 mM $Na^+$ (G), or 70 mM $Ca^{2+}$ (H), showing PC1-TLD/PC2_AA gave rise to current with similar properties as full-length PC1/PC2_AA channel.

I   Co-IP followed by Western blot showing the association between PC2_AA and the indicated full-length or fragment PC1. Bands of full-length (asterisk), GPS-cleaved CTF or expressed CTF fragment (open circle), and TLD (star) of PC1 are indicated.

J   Surface biotinylation followed by Western blot showing the expression of the complexes formed between PC2_AA and full-length PC1, PC1-CTF, or PC1-TLD at the oocyte surface. Besides the blots shown in the figure, surface samples were also blotted with an anti-PC1 N-terminus antibody, and the cleaved NTF fragment was found associated with the plasma membrane in the full-length sample (shown in Appendix Fig S4). Bands of full-length (asterisk), GPS-cleaved CTF or expressed CTF fragment (open circle), and TLD (star) of PC1 are indicated.

Source data are available online for this figure.

channel (Fig 4C). Furthermore, its current in 70 mM $Ca^{2+}$ solution is also similar to that of the full-length channel (Fig 4D), and the current has the CaCC component which can be inhibited by MONNA (Appendix Fig S3A). These results indicate that the channel function of the PC1-CTF/PC2_AA complex resembles that of full-length PC1/PC2_AA. Therefore, the CTF is sufficient for forming a functional channel with PC2 (shown with PC2_AA here). Coexpressing PC1-CTF with WT PC2, or introducing a mutation in the putative pore region of PC1-CTF, R4090W, completely abolished the current of PC1-CTF/PC2_AA (Fig 4E), which strongly supports the role of PC1-CTF in conducting currents together with PC2_AA and rules out the likelihood that currents be mediated by induced endogenous channels.

We then sought the PC1 core channel functional unit by further shortening the fragment. The similarity between PC2 and the S6-S11 of PC1 suggests a potential ion channel role of the latter. Indeed, the cryo-EM structure of PC1/PC2 indicates that the PC1 S6-S11 assemble with three PC2 proteins into a TRP channel-like structure [42]. Interestingly, a natural PC1 cleavage product, P100, has been identified and likely contains S6-S11 and the intracellular C-terminal tail [57], suggesting a physiological role of this part of PC1, which we named PC1 TRP-like domain (TLD) here. We tested the channel function of PC1-TLD by generating a construct including the fragment from G3592 to T4293 of PC1, which is composed of the loop between S5 and S6, the transmembrane domains S6 to S11, and the intracellular C-terminal tail (Fig 4A). The results show that coexpressing PC1-TLD with PC2_AA produced a functional channel. Although the currents of PC1-TLD/PC2_AA are much smaller, it resembles the current of the full-length channel in all tested solutions (Fig 4F–H). PC1-TLD/PC2_AA channel is not blocked by 2 mM $Ca^{2+}$ (Fig 4F). In the 70 mM $Ca^{2+}$ solution, the trough at strong hyperpolarization of the current of PC1-TLD/PC2_AA (Fig 4H) and the inhibition of the current by MONNA (Appendix Fig S3B) shows the activation of CaCC, indicating the $Ca^{2+}$ permeability of this channel. Consistently, co-IP and surface biotinylation experiments showed that, as full-length PC1, PC1-CTF and PC1-TLD can associate with PC2_AA, and the resulting complexes traffic to the plasma membrane (Fig 4I and J, Appendix Fig S4). Thus, our data show that PC1-CTF and even PC1-TLD are sufficient for ion channel function of PC1 in the PC1/PC2 complex.

In most of our experiments, we could only record small currents of PC1-TLD/P2C_AA, which suggests a potential role of the S1-S5 domains in facilitating PC1 channel activity. At the same time, we also observed less surface PC2_AA when it was coexpressed with PC1-LTD than when it was with either of the two longer PC1 fragments (Fig 4J). Thus, missing S1-S5 domains may have induced a partial defect in channel folding or trafficking of the PC1/PC2 complex. Further study is needed to investigate the roles of S1-S5 in the function of PC1.

As negative controls, expressing PC1, PC1-CTF, or PC1-TLD alone did not give rise to significant currents in a solution containing 100 mM $Na^+$ and 2 mM $Ca^{2+}$ (Fig EV3). We occasionally noticed that expression of PC1-CTF alone, but not the full-length PC1 or PC1-TLD, led to tiny whole-oocyte currents which are much smaller than that when PC2_AA was coexpressed (Fig EV3). It will be interesting to figure out whether this current is from a complex formed by PC1-CTF with endogenously expressed TRPP or other proteins.

**PC1/PC2 complex channel has different pore properties compared to the homomeric PC2 channel**

Both the fact that PC1/PC2_AA has higher $Ca^{2+}$ permeability than PC2_AA (Fig 3) and a mutation (R4090W) in the putative pore region of PC1 can abolish the PC1-CTF/PC2_AA current (Fig 4E) suggest that PC1 contributes to the channel pore. Clearly, this indicates that the pore of the complex channel is different from that of the homomeric PC2 channel. To confirm this, we measured the permeability of the two channels to several inorganic and organic cations by analyzing the RP of the I–V curves. Solutions containing 100 mM of the following ions were tested: $Na^+$ [atomic mass $(m_a) = 23$], $Li^+$ $(m_a = 7)$, $Cs^+$ $(m_a = 133)$, dimethylamine$^+$ ($DMA^+$, molecular weight (MW) = 45.1), diethylamine$^+$ ($DEA^+$, MW = 73.1), tetraethylammonium$^+$ ($TEA^+$, MW = 130.3), and N-methyl-D-glucamine$^+$ ($NMDG^+$, MW = 195.2). Since the majority of the cations inside the oocytes are $K^+$, the RP of the current recorded in the solution of a tested ion reflects the permeability ratio (selectivity) between the tested ion and $K^+$. A more positive RP means the channel tends to be more selective to the tested ion. The results show that the RPs of PC2_AA currents in these ions can be

divided into two groups. All inorganic ions fall into the range from −15 mV to −25 mV, while all organic ions fall into a much more negative range from −55 to −80 mV (Fig 5A, red I–V curves), indicating that the PC2_AA channel is much less permeable to the larger organic ions. Previously, PC2 has been shown to have good permeability to DMA, TEA, and even larger organic ions [58]. Our results may reflect the difference between the WT and the GOF PC2_AA channels. When PC1-CTF/PC2_AA current was recorded with these

ions, dramatic RP changes were observed. The RPs in all ions, except for NMDG, became much more positive (Fig 5A, blue I–V curves, and B). The RPs of the PC1-CTF/PC2_AA channel current in $DMA^+$ and $DEA^+$, as two extreme examples, shifted 80.2 mV and 63.2 mV, respectively, in comparison with that of the PC2_AA channel current (Fig 5A and B).

To gain more insights into difference in pore properties between these two channels, we calculated the permeability ratio between

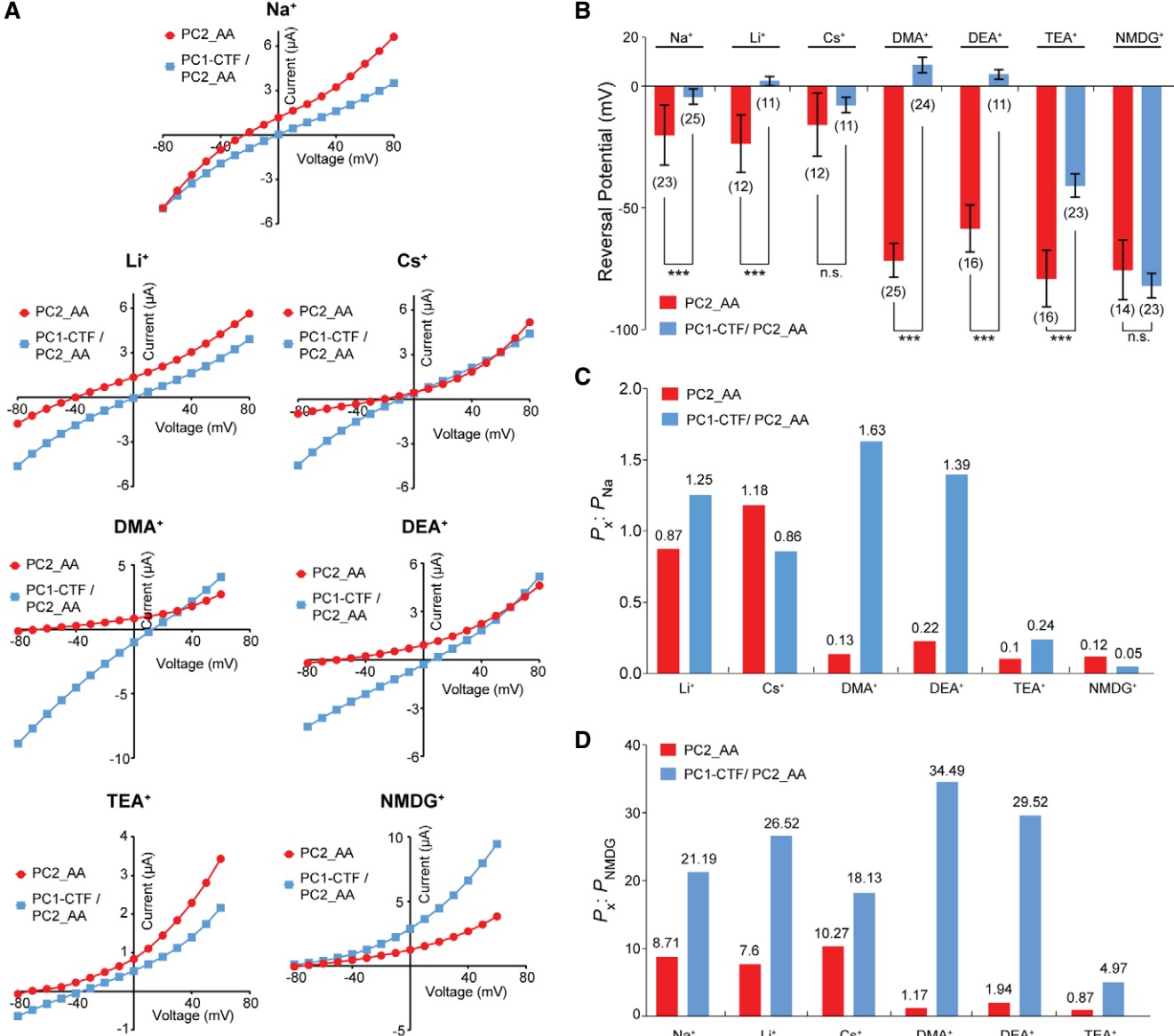

**Figure 5. PC1-CTF/PC2_AA heteromeric channel has a different ion permeability to that of the PC2_AA homomeric channel.**

A   Representative I–V curves of the PC2_AA and PC1-CTF/PC2_AA channels in divalent ion-free bath solutions containing 100 mM of indicated ions, showing the differences in reversal potential.

B   Bar graphs showing the markedly different reversal potentials of PC2_AA channel and the PC1-CTF/PC2_AA channel in bath solutions containing 100 mM of the indicated ions. Oocyte numbers are indicated in parentheses. Data are presented as mean ± SD (n.s.: not significant, ***P < 0.001, Student's t-test).

C, D   Bar graphs showing the permeability ratios of the indicated ions to that of $Na^+$ ($P_x$: $P_{Na}$) (C) and $NMDG^+$ ($P_x$: $P_{NMDG}$) (D). These ratios are strikingly different between the two channels. The ratios of the particular ions, which are the values of the bars, are indicted on top of the bars.

Source data are available online for this figure.

different ions tested. Since the exact intracellular $K^+$ concentration is unknown, we are not able to accurately calculate the permeability ratios of the tested ions to that of $K^+$. Instead, we used $Na^+$ as a reference ion and calculated the permeability ratios of the other tested ions (x) to it ($P_x/P_{Na}$) using the Goldman–Hodgkin–Katz equation [59]. The results show that compared to PC2_AA, the PC1/PC2_AA channel has greater $P_x/P_{Na}$ of $Li^+$, $DMA^+$, $DEA^+$, and $TEA^+$, and smaller $P_x/P_{Na}$ of $Cs^+$ and $NMDG^+$ (Fig 5C). The largest ratio increase was seen for $DMA^+$ and $DEA^+$, indicating a significantly greater permeability of the PC1/PC2_AA channel to these large ions. Next, we calculated the permeability ratios of the tested ions to $NMDG^+$ ($P_x/P_{NMDG}$). Since the currents of the two channels have similar RPs in $NMDG^+$, compared to $Na^+$, $NMDG^+$ serves as a better reference ion to evaluate the difference between the pores of the two channels. As expected, the $P_x/P_{NMDG}$ of all ions for PC1-CTF/PC2_AA are remarkably greater than those of the PC2_AA (Fig 5D). These results show that the PC1-CTF/PC2_AA channel has greater permeability to these ions, indicating it has a larger or more flexible ion-conducting pore than the PC2_AA channel.

Another piece of evidence showing the pore difference between the PC1/PC2 and PC2 channels came from measuring the effects of channel blockers. Previously, we found that the cation channel blockers $Gd^{3+}$, ruthenium red (RuR), and amiloride dramatically block the PC2_F604P channel [22]. In the current experiments, these three blockers have significantly different blocking effects on the PC2_AA and PC1-CTF/PC2_AA channels. $Gd^{3+}$ at 0.5 mM, but not 0.1 mM, greatly blocked the inward current of PC2_AA channel (by 93% at −80 mV) (Fig EV4A). However, 0.5 mM $Gd^{3+}$ only blocked 48% of PC1-CTF/PC2_AA channel at −80 mV (Fig EV4A). RuR at 0.01 and 0.1 mM blocked 68 and 75% of PC2_AA channel at −80 mV, respectively, but only 35 and 43% of PC1-CTF/PC2_AA channel (Fig EV4B). In contrast, amiloride blocked more the PC1-CTF/PC2_AA channel than PC2_AA channel. Amiloride at 1 and 5 mM blocked 13 and 48%, respectively, of PC2_AA channel current at −80 mV, while it blocked 48 and 83%, respectively, of PC1-CTF/PC2_AA channel current at the same voltage (Fig EV4C). Since these blockers most likely function by binding to the channel pore, these results further supported that the pores of the two channels are quite different.

The above findings demonstrate that PC1-CTF/PC2_AA has a different channel pore from PC2_AA, and suggest that PC1-CTF directly participates in forming the ion-conducting pore of the complex. However, an alternative explanation would be that the differences found here reflect a conformational change in the pore of the homotetrameric PC2_AA channel caused by the association with PC1-CTF. We can rule out this possibility if pore mutations in both proteins can change ion permeability of the complex channel.

## Pore mutations of PC1 and PC2 change ion selectivity of the PC1-CTF/PC2_AA channel

A widely held and extensively tested prediction of channel-forming proteins is that mutating residues lining the ion conduction pathway significantly alters ion selectivity (not simply current amplitude) of the channel. Based on the cryo-EM structures of PC2 and PC1/PC2 complex [20,42], we selected several amino acids at the pore regions of both PC1 and PC2, and tested how mutations at these sites affect ion selectivity (Fig 6A).

Three amino acid residues, L641, G642, and D643, form the ion selectivity filter of the homomeric PC2 channel [20]. In the reported PC1/PC2 structure, they are also located at the putative ion selectivity filter position (Fig 6A) [42]. While mutating the hydrophobic L641 to negatively charged aspartic acid (D) did not significantly alter RP of currents of two large ions, $TEA^+$ and $NMDG^+$, it dramatically shifted the RP in $Na^+$, $Li^+$, $Cs^+$, $DMA^+$, and $DEA^+$ toward negative voltages, indicating the reduction in the permeability ratios between the tested ions and $K^+$ (Fig 6B for RP changes and Fig EV5 for representative I–V curves). The D643N mutation also led to significant, though mild, negative shifts of the RP in $Li^+$, $DMA^+$, and $DEA^+$ (Figs 6C and EV5). We also mutated N674 on S6 of PC2, which is located directly above the lower gate in the PC2 homomeric channel [20], and its side chain faces into the pore of the PC1/PC2 channel (Fig 6A) [42]. The N674C mutation, similar to L641D, led to dramatic negative RP shift for all ions tested except for $NMDG^+$ (Figs 6B and EV5). Accordingly, for most tested ions, the permeability ratios $P_x/P_{Na}$ and $P_x/P_{NMDG}$ of these mutants, especially L641D and N674C, are dramatically altered (Fig 6D and E). For example, for L641D, $P_{Li}/P_{Na}$, $P_{DMA}/P_{Na}$, and $P_{DEA}/P_{Na}$ are all smaller, while $P_{TEA}/P_{Na}$ and $P_{NMDG}/P_{Na}$ are both greater than that of the pseudo-WT channel (Fig 6D). At the same time, $P_x/P_{NMDG}$ of all tested ions, except for $TEA^+$, are substantially smaller than that of the pseudo-WT channel (Fig 6E). These data confirm that PC2 directly participates in the formation of the channel pore of the PC1/PC2 channel.

---

**Figure 6. Pore mutations in PC1 or PC2 led to changes in ion permeability of the PC1-CTF/PC2_AA channel.**

A    TOP: a bottom view of a previously reported cryo-EM structure of PC1/PC2 channel (PDB code 6A70) [42], which is missing the NTF of PC1 and the intracellular C-terminal tails from both proteins. S10-S11 of one PC1 subunit and S5-S6 from three PC2 subunits assembled to form the pore. Bottom: The mutated amino acids in this experiment (in purple) are mapped on S10-S11 of one PC1 and S5-S6 of one PC2 subunit in the complex. The dashed gray arrow indicates the putative ion flow path. Since human PC1 was used for structure determination, mouse amino acids mutated in this study are indicated in parentheses. Due to the low resolution of the PC1 pore region in the structure, side chains are not seen for the three mutated amino acids at the putative selectivity filter region. E4068 (E4078 in human) is not solved in the structure, and it is labeled to indicate its approximate location in the structure.

B, C    Bar graphs showing the reversal potentials of PC1-CTF/PC2_AA and the indicated mutants tested in bath solutions containing 100 mM of the indicated ions. Two PC1 and two PC2 mutations caused dramatic changes in the reversal potential for almost all tested ions (B), indicating that these amino acids are essential for ion permeability. Another two PC1 mutations and one PC2 mutation only led to relatively small changes of reversal potential for some ions but not others (C), indicating a less important role of these amino acids in ion permeability. Statistical significance between reversal potentials of all pore mutants and that of the PC1-CTF/PC2_AA channel is indicated. Representative I–V curves are shown in Fig EV5. Oocyte numbers are indicated in parentheses. Data are presented as mean ± SD (n.s.: not significant, **$P$ < 0.01, ***$P$ < 0.001, Student's $t$-test).

D, E    Bar graphs showing the permeability ratios of the indicated ions to that of $Na^+$ ($P_x$: $P_{Na}$) (D) and $NMDG^+$ ($P_x$: $P_{NMDG}$) (E). Pore mutations in both PC1 and PC2 lead to significant changes in the permeability ratios in most of the cases.

Source data are available online for this figure.

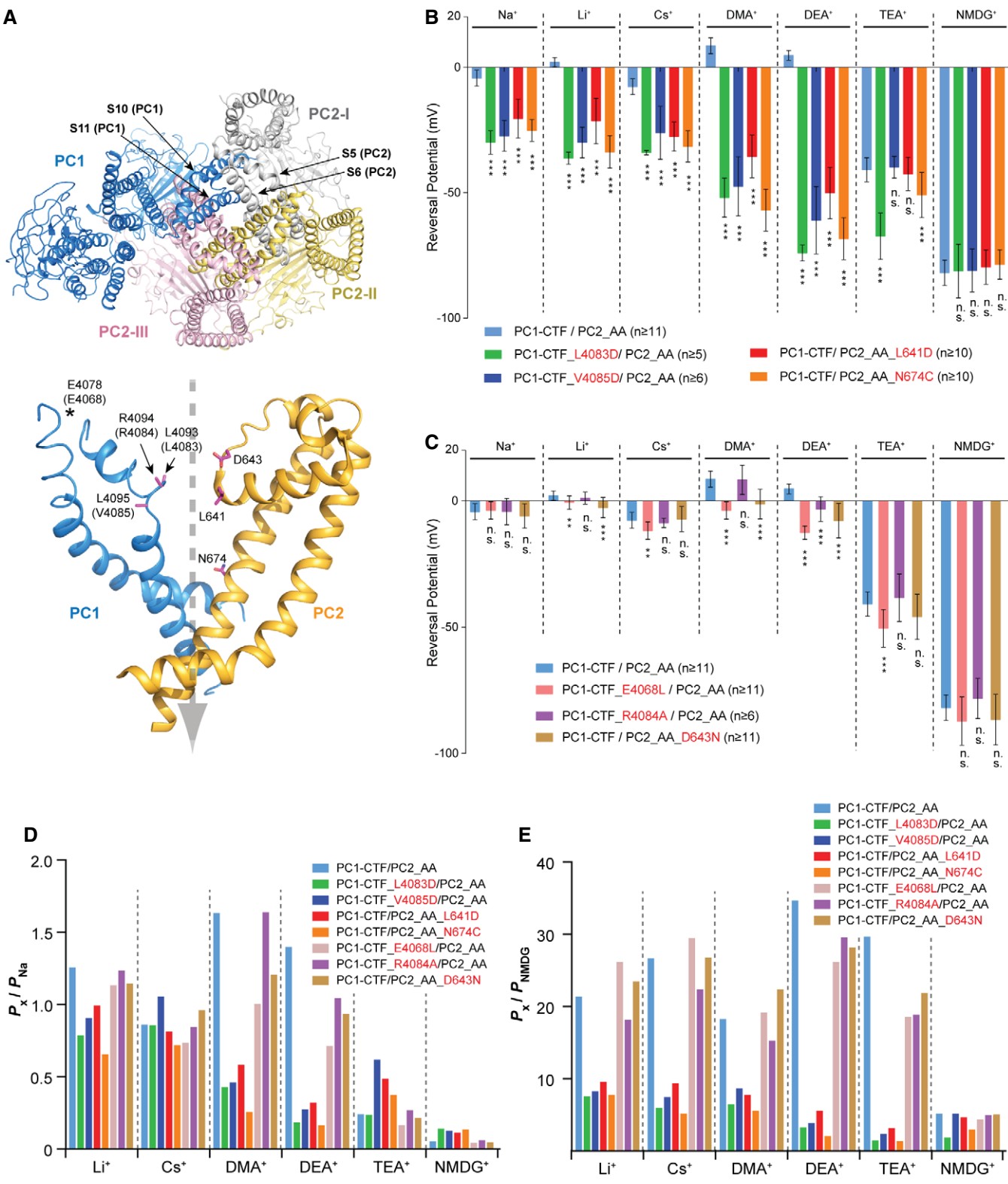

Figure 6.

In the published cryo-EM structure of PC1/PC2, the conformations of the pore region of PC1 are very different from that of PC2 [42]. The S6 of PC1 is bent in the middle, and the selectivity filter and canonical TRP pore helices are missing (Fig 6A) [42]. Protein flexibility leads to low resolution in cryo-EM structures, which may have contributed to the poor resolution of the top of the PC1 pore region and the linker between S5 and S6 [42]. As a consequence, side chains of many residues in this region, including those aligned with

the selectivity filter of PC2, L4093, R4094, and L4095 (L4083, R4084, and V4085 in mouse PC1), are not seen in the structure (Fig 6A). Mutation of mouse PC1 L4083 or V4085 to aspartic acid produced similarly drastic effects on ion selectivity as the L641D and N674C mutations did in PC2. L4803D led to a dramatic negative RP shift in all tested ions except for NMDG$^+$ (Figs 6B and EV5). The mutant channel has smaller $P_{Li}/P_{Na}$, $P_{DMA}/P_{Na}$, and $P_{DEA}/P_{Na}$, and larger $P_{NMDG}/P_{Na}$, as well as smaller $P_x/P_{NMDG}$ of all tested ions than pseudo-WT channel (Fig 6D and E). V4085D caused similar negative shifts in RP except for that in TEA$^+$ and NMDG$^+$ (Figs 6B and EV5). The mutant channel has larger $P_{Cs}/P_{Na}$ and $P_{TEA}/P_{Na}$, smaller $P_x/P_{Na}$ of other ions, and smaller $P_x/P_{NMDG}$ of almost all ions than pseudo-WT channel (Fig 6D and E). Interestingly, the mutation R4084A, located between the two mutations above, only caused a small negative RP shift and relative permeability reduction in DEA$^+$ and did not significantly affect the permeability to the other ions (Figs 6C–E

and EV5). E4068, the closest negatively charged amino acid to the putative selectivity filter in the PC1/PC2 structure, is located in the unsolved part of the S5-S6 linker of PC1 (Fig 6A). When we mutated it to hydrophobic leucine, the RP in Li$^+$, Cs$^+$, DMA$^+$, DEA$^+$, and TEA$^+$ was all significantly shifted toward negative voltage, suggesting this amino acid contributes to ion permeability (Figs 6C–E and EV5). These data strongly indicate that the PC1 subunit directly participates in forming the channel pore together with PC2 and that the region L4083-V4085 plays a crucial role in the ion selectivity, as suggested by the cryo-EM structure.

### GPS cleavage is not necessary for the GOF PC1/PC2_AA channel activity

PC1/PC2_AA has many potential applications in studying the function and regulation of this channel. To illustrate this, we studied

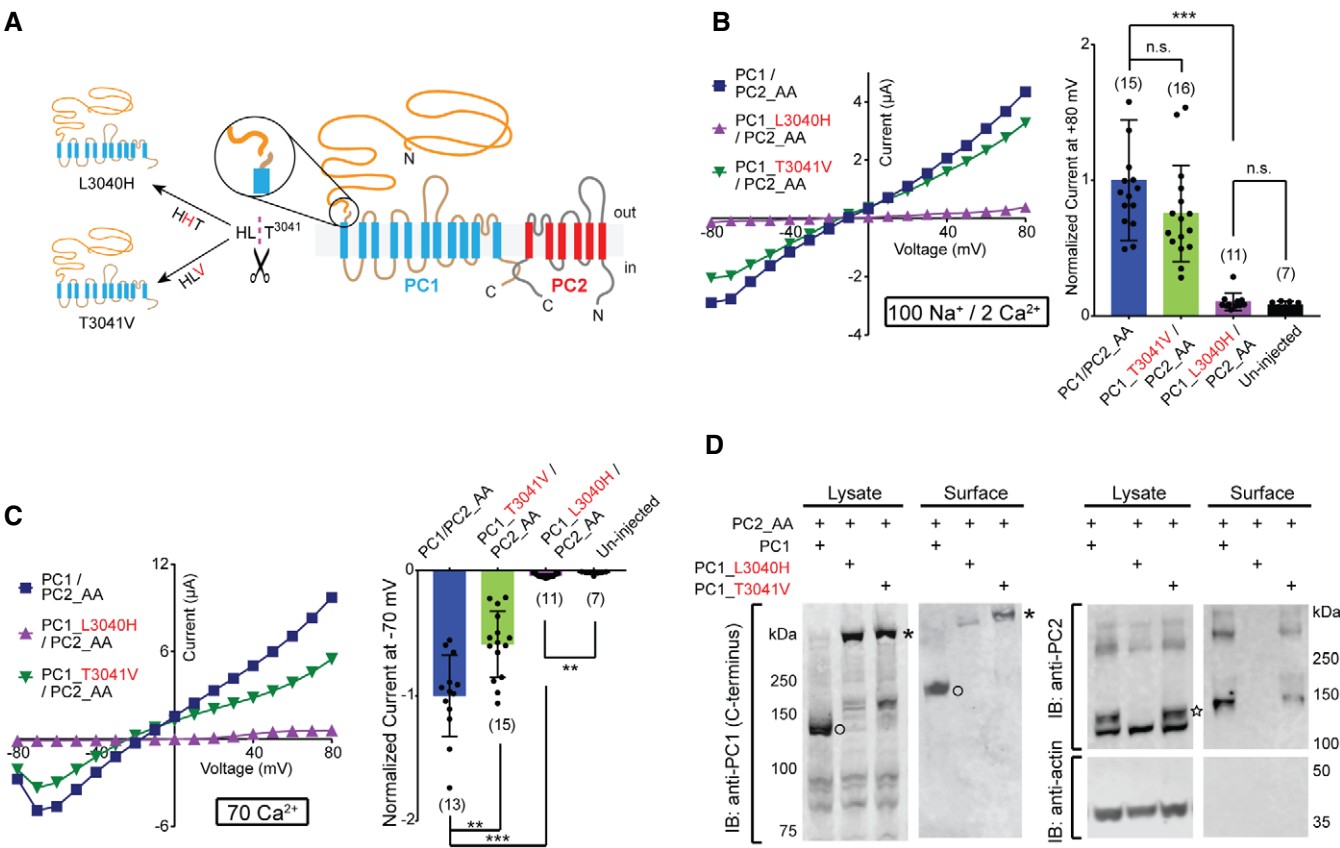

**Figure 7. GPS cleavage of PC1 is not necessary for channel activity of the PC1/PC2_AA channel.**

A    Transmembrane topology of PC1 and PC2 proteins, showing the cleavage at the N-terminal GPS site (in circles), the "HLT" tripeptide where the cleavage happens, and the two mutations, L3040H and T3041V, that abolish GPS cleavage.

B, C  Representative currents and scatter plot and bar graphs showing the currents of the full-length WT and mutant PC1s associated with PC2_AA in bath solutions containing 100 mM Na$^+$ and 2 mM Ca$^{2+}$ (B) or 70 mM Ca$^{2+}$ (C). In scatter plot and bar graphs, currents were normalized to the average current of PC1/PC2_AA recorded from the same batch of oocytes. Oocyte numbers in scatter plot and bar graphs are indicated in parentheses. Data are presented as mean ± SD in bar graphs (n.s.: not significant, **$P$ < 0.01, ***$P$ < 0.001, Student's $t$-test).

D    Surface biotinylation followed by Western blot showing the surface expression of the PC1_T3041V/PC2_AA complex, but not the PC1_L3040H/PC2_AA complex, in *Xenopus* oocytes. PC2_AA is completely absent from the surface sample when PC1_L3040H was coexpressed, indicating that, in our experimental condition, almost all PC2_AA were in the complex with PC1_L3040H and trapped in the process of cell surface trafficking. Bands of full-length (asterisk) and GPS-cleaved CTF (open circle) of PC1 and 130 kDa PC2 (star) are indicated.

Source data are available online for this figure.

whether N-terminal GPS cleavage is necessary for the channel activity of the PC1/PC2_AA channel. GPS cleavage occurs at the HL*T$^{3041}$ tripeptide (*: scissile bond. Amino acids are numbered as in mouse PC1) [31] (Fig 7A). Two mutants that abolish the cleavage of PC1, L3040H and T3041V, have been found to have defective PC1 *in vivo* functions in mice. PC1_L3040H appears to have complete loss of the function since *PC1-L3040H-BAC* transgenic mouse lines did not rescue the embryonic lethality of *Pkd1*$^{-/-}$ mice [32]. However, PC1_T3041V exhibited partial PC1 function since the knock-in mice escaped embryonic lethality and had normal-appearing kidneys at birth, although they developed rapid renal cysts in distal nephron segments later [29]. In order to understand the effect of the GPS cleavage on the PC1/PC2 channel function, and the functional difference between the two non-cleavable mutants, we tested the mutants with the GOF PC1/PC2 channel.

Coexpression of PC1_T3041V and PC2_AA in oocytes gave similar channel currents to that of PC1/PC2_AA, although the current of PC1_T3041V/PC2_AA is slightly smaller in 70 mM Ca$^{2+}$ (Fig 7B and C). However, no channel current was recorded when PC1_L3040H was coexpressed with PC2_AA (Fig 7B and C). We then checked the protein expression and plasma membrane trafficking. Both L3040H and T3041V mutants expressed well and were not cleaved since only full-length proteins were seen in Western blot (Fig 7D, the image on the left). However, surface biotinylation revealed that PC1_T3041V/PC2_AA expressed robustly on the oocyte plasma membrane, while PC1_L3040H/PC2_AA did not traffic to the plasma membrane at all since coexpressed PC2 was not detectable in the surface sample (Fig 7D). This result is consistent with the previous finding that PC1_T3041V acquires EndoH resistance, implying it can at least traffic to Golgi [45], while PC1_L3040H is stuck in the ER and remains entirely EndoH-sensitive [32]. Consistently, the 130 kDa band of PC2 was not seen when PC1_L3040H was coexpressed (Fig 7D). The fact that no PC2 was detected on the surface when PC1_L3040H was coexpressed also suggests that under our experimental condition, the majority of PC2 is in complex with PC1 and that almost no homomeric PC2 channel exists.

Thus, our results suggest that GPS cleavage is not necessary for PC1 to play its channel role in complex with PC2_AA as long as the complex traffics to the plasma membrane. However, since the studied GOF channel is constitutively open, we could not test whether the GPS cleavage plays a role in channel gating.

## Discussion

In this study, for the first time, we were able to generate a GOF PC1/PC2 channel and dissect the role of PC1 in this complex. Although channel current mediated by the PC1/PC2 complex has been previously reported in several experimental systems [9,60,61], replication has been inconsistent [20,40]. In the current study, we generated several new GOF PC2 mutant channels by changing residues at the lower gate and found one double mutation, L677A/N681A (AA), that led to the opening of the PC1/PC2 channel. The new heteromeric GOF channel has significantly different permeability to Ca$^{2+}$ (Fig 3) and monovalent cations (Fig 5) and different responses to divalent cations (Fig 3 and Appendix Fig S2) and channel blockers (Fig EV4), compared with the PC2_AA homomeric channel. More importantly, mutations in either PC1 or PC2 pore region can change this channel's

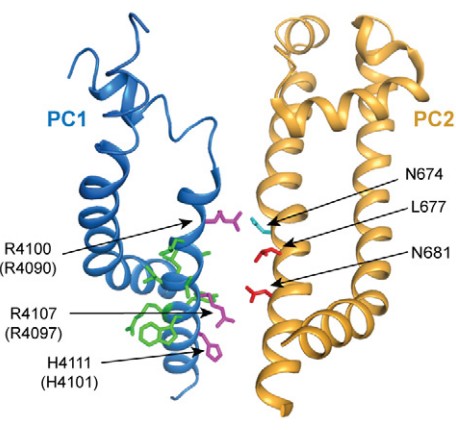

**Figure 8. Several key residues in the pore of the PC1/PC2 channel may play roles in ion conductance.**

Structure of the PC1/PC2 complex showing the side view of S10-S11 from the PC1 subunit and S5-S6 from one PC2 subunit in the cryo-EM structure of PC1/PC2 (PDB code 6A70) [42]. Side chains of PC2-L677A and N681 are shown in red and N674 in cyan. Side chains of amino acids on S11 of PC1 at this region were shown in purple for the three positively charged amino acids and in green for others. In PC1/PC2 complex, no bulky side chain from PC1 lines up with PC2-L677A and N681 to form the restriction at this position, while R4090 of PC1 and N674 of PC2 narrow down the channel pore in the middle.

ion permeability (Fig 6). All these data confirm that the currents we recorded are conducted by PC1/PC2_AA heteromeric channel.

Association with PC1 leads to distinct effects on PC2 GOF mutants, F604P and AA. PC1/PC2_AA produced a robust current, while the PC1/PC2_F604P current was much smaller. This difference should be due, at least in part, to the fact that the two mutations open the channel pore in different ways. Mutation F604P abolishes the hydrophobic interaction between S5 and S6, which leads to twisting and rotation of the four PC2 S6 helices to open the lower gate of the homomeric channel [21,43]. When in complex with PC1, it is possible that either the F604P-induced S6 conformational changes are somehow blocked by the PC1 subunit, or the PC1 subunit directly clogs the pore, resulting in a smaller conductance of PC1/PC2_F604P in comparison with homomeric PC2_F604P (Fig EV1). In the case of PC2_AA, the alanine mutations at L677 and N681 shorten the side chains that form the restriction at the lower gate, directly widening the pore size, and therefore, the channel's overall conformation would remain in a closed state, as the previous study indicated [43]. Thus, PC2_AA is distinct from PC2_F604P with respect to the way by which the channel pore is opened. The conductance of their opened pore is also different, which is supported by the difference in current size (Fig 2B and C) and the larger Ca$^{2+}$ permeability of PC2_AA (Fig 2D). These results also suggest that the lower gate of the channel pore plays a role in defining the Ca$^{2+}$ permeability in PC2. Similarly, it was found that mutations at the inner pore/lower gate region in TRPA1 affect permeability of ions, including Ca$^{2+}$ [62]. In the case of the PC1/PC2_AA complex, the substitution of one of the four PC2_AA subunits with the PC1 subunit presumably is not sufficient to block the heteromeric lower pore, allowing constitutive cation permeation. Indeed, in the cryo-EM structure of PC1/PC2 [42], there is no bulky side chain from the

PC1 subunit that lines up with PC2-L677 and N681 (Fig 8). AA mutations on three PC2 subunits are sufficient to open up the restriction at this position.

The cryo-EM structure of PC1/PC2 shows that three positively charged amino acids on S11 of PC1, R4100, R4107, and H4111 (align to R4090, R4097, and H4101 in mouse PC1) face to the pore and are proposed to disfavor cation penetration (Fig 8) [42]. This feature suggests a possibility that the solved structure may be in a closed state, and these residues will be moved out of the pore when the channel is gated. In the GOF PC1/PC2_AA channel, its overall conformation most likely remains in a closed state, and these three positively charged residues on PC1 should still be in the pore. Since we can record robust current, at least in this GOF channel, ions can pass through in spite of having these positive charges in the pore. Among the three residues, only R4090 is located above the gate residues L677 and N681 on PC2. It pairs up with N674 in the PC2 subunits and narrows down the pore (Fig 8). Indeed, mutation R4090W, which adds an even bulkier side chain at this position, completely abolished the PC1-CTF/PC2 current (Fig 4E). At the same time, mutation PC2-N674C greatly changed the ion permeability of the channel (Fig 6B). These results indicate a crucial role of the R4090-N674 position in regulating ion permeability of PC1/PC2.

The presence of PC1 in the channel complex significantly alters channel properties, including ion selectivity. Firstly, common cation channel blockers inhibit the PC1/PC2_AA and PC2_AA channels differently (Fig EV4). Secondly, both extracellular 2 mM $Ca^{2+}$ and $Mg^{2+}$ block the PC2_AA channel but not the PC1/PC2_AA channel (Figs 3B and Appendix Fig S2), and PC1/PC2_AA has a relatively larger $Ca^{2+}$ permeability than PC2_AA (Fig 3D and F). While these data, obtained through GOF mutants, may not reflect what is really happening in the WT channels, the $^{45}Ca$ uptake experiment directly shows that the WT PC1/PC2 channel indeed has higher $Ca^{2+}$ permeability than PC2 alone (Fig 3G and H). Thirdly, our results show that the PC1/PC2_AA channel has a larger or more flexible pore, and higher permeability to most monovalent cations tested in this study (i.e., $Na^+$, $Li^+$, $Cs^+$, $DMA^+$, $DEA^+$, and $TEA^+$) when compared to PC2_AA alone (Fig 5). This is consistent with the structure revealed by the cryo-EM, where the upper pore region of PC1 is more flexible due to missing pore helices (Fig 6A) [42]. Collectively, these results tell us that the heteromeric PC1/PC2 channel and the homomeric PC2 channel are two distinct channels that likely have their unique functions. The fact that the homomeric PC2 channel has low $Ca^{2+}$ permeability [11,22] while the PC1/PC2 channel complex has much higher $Ca^{2+}$ permeability indicates their very distinct physiological roles and thus contributions to ADPKD.

Recently, two groups have reported PC2-dependent channel activity on primary cilia by doing single-channel recording directly from cilia [11,16]. The currents recorded in the two studies are significantly different on $Ca^{2+}$ permeability, which has been extensively discussed [11]. Briefly, the $Ca^{2+}$ permeability of the current reported in Kleene and Kleene [16] is more than 20 times higher than that of the current reported in Liu et al [11]. The latter found that their current is PC1-independent and it has similar ion permeability and extracellular $Ca^{2+}$-blocking as that of the GOF PC2_F604P [11], which indicates that their current is most likely conducted by the homomeric PC2 channel. It will be very interesting to find out if the current recorded by Kleene and Kleene [16] is conducted by the higher $Ca^{2+}$-permeable PC1/PC2 complex instead. In our recordings, to get significant

$Ca^{2+}$ influx, we needed to clamp the oocytes at a hyperpolarized negative membrane potentials and apply a high extracellular concentration of $Ca^{2+}$ (Fig 3F). A previous study determined that the resting membrane potential of cilia is about $-18$ mV [12]. If the naturally gated WT PC1/PC2 channel has a similar $Ca^{2+}$ permeability as the GOF PC1/PC2_AA channel, then under this membrane potential and relatively low physiological $Ca^{2+}$ concentrations, we would expect a small $Ca^{2+}$ influx through this channel upon opening by unknown stimuli. However, due to the extremely small volume of cilia [12], the $Ca^{2+}$ influx through the PC1/PC2 channel might be sufficient to trigger $Ca^{2+}$ signaling in cilia. At the same time, we cannot rule out the possibility that the naturally gated PC1/PC2 channel has greater $Ca^{2+}$ permeability than the GOF channel.

Our results showed that the PC1-CTF preserves almost the same channel function as that of the full-length protein in complex with PC2 (Fig 4B–D). The CTF is a naturally occurring PC1 fragment generated by GPS cleavage and usually remains attached to the cleaved NTF through non-covalent interaction [30,63]. However, detachment does occur [56]. In some aGPCRs, NTF and CTF were found to function as separate proteins in cell surface reception and signaling [64]. In the case of PC1, the detached CTF may also have NTF-independent channel function when assembled with PC2 (Fig 4). The cryo-EM structure of PC1/PC2 was solved using a truncated PC1 fragment containing the CTF lacking the intracellular C-terminal tail [44]. The structure shows that this PC1 fragment assembles with PC2 into a channel-like complex [44], consistent with our functional data. We further show that the shorter fragment, TLD, retains channel function when associated with PC2 (Fig 4F–H). Together with the structural data [44], this result shows that TLD is the core structural component for PC1's channel function. The cleaved P100 fragment of PC1, which was found in mouse kidney [57], should be very similar to TLD in sequence. Our study, therefore, infers that this naturally cleaved fragment of PC1 may have its own physiological function.

Although essential for ciliary trafficking of PC1 [45], GPS cleavage is not necessary for PC1 channel function in the PC1/PC2_AA complex in our experimental system, since the non-cleavable PC1_T3041V contributed full channel activity (Fig 7). One of the reasons that PC1_T3041V is pathogenic in mouse is that it cannot traffic to primary cilia [45]. Our conclusion is consistent with the observation that PC1_T3041V retains function to some extent since knock-in mice exhibited milder cystic phenotypes than PC1 null mice [29]. Indeed, a significant portion of PC1 remains as non-cleaved full-length protein in the embryonic kidney, indicating a special function of this form in embryonic development [29,65]. We show that as long as this non-cleaved form reaches the plasma membrane in complex with PC2, it will contribute to channel function. Together with the function of PC1-CTF, these results suggest that the presence or absence of NTF has little influence on the GOF PC1/PC2 channel function. However, because we only examined the effect on the GOF channel, which is constitutively open, the potential roles of GPS cleavage and the NTF in ligand binding and channel gating in WT channel remain to be determined. In aGPCRs, cleaved NTF was found associated with the membrane-tethering CTF and may function as an endogenous ligand, or be involved in ligand binding [66]. It will be valuable to assess whether PC1-NTF also plays a role in modulating PC1/PC2 channel function.

Studying the role of PC1 in the PC1/PC2 complex is very challenging due to the lack of a known activation mechanism. The GOF

channel generated here provides a platform for the study of the function and regulation of this complex. With our GOF mutant, we were able to reliably record the ion channel activity of this complex and dissect the function of PC1. By making mutations in the pore region, we provide direct evidence that the PC1 protein functions as an ion channel pore-forming subunit in the PC1/PC2 complex. In combination, the results from this study with our previous finding of the 1:3 subunit stoichiometry [40], and the recently published cryo-EM structure [44], have significantly advanced the understanding of channel function of PC1/PC2.

# Materials and Methods

### cDNA constructs and cloning

Full-length mouse PC1 cDNA (NCBI accession No. NM_013630) and human PC2 cDNA (NCBI accession No. U50928) were cloned into a modified pGEMHE vector for *in vitro* transcription. The signal peptide (first 23 amino acids in the N-terminus) of PC1 was replaced by an Ig k-chain leader sequence (from the pDisplay vector from Invitrogen), and a FLAG tag was added immediately after this sequence. The PC2 used in this study has an N-terminal HA tag. For CTF and TLD of PC1, the cDNAs encoding the protein fragment from T3041 to the C-terminal end (T4293) and the fragment G3592-T4293, respectively, were cloned into the modified pGEMHE vector. An Ig k-chain leader sequence was added to the N-terminus of the CTF construct, and both CTF and TLD contain a FLAG tag on the N-terminus. All mutations were generated with PCR and confirmed by sequencing.

### Electrophysiology

Whole-oocyte currents of *Xenopus laevis* oocytes were recorded with the two-electrode voltage clamp (TEVC) technique. All recordings have been repeated with at least two batches of oocytes, and most of them have been done three or more times. RNAs were synthesized *in vitro* and injected into follicle-membrane-free *Xenopus* oocytes. For every oocyte, 30 ng of PC2 was injected. When injecting the full-length PC1 and PC2 combination, the quantity of injected PC1 RNA is 1.5 times in the molar ratio of PC2. This is increased to 2 times in the case of PC1-CTF and PC1-TLD. The excess amount of PC1 protein is to eliminate the homomeric PC2 complex formation. For oocytes expressing PDK1-CTF alone, 100 ng of PC1-CTF was injected per oocyte.

After injection, the oocytes were incubated at 18°C before whole-oocyte currents were recorded. The incubation time is 2–3 days for oocytes injected with PC2 RNA alone, PC1_CTF RNA alone, combined PC1-CTF and PC2_AA RNAs, combined PC1-TLD and PC2_AA RNAs, and their corresponding mutants. The incubation time is 3–5 days for oocytes injected with combined full-length PC1 and PC2_AA RNAs and their corresponding GPS mutants. The longer incubation time is necessary for full-length PC1-including complex due to its slower trafficking to the plasma membrane.

### Electrophysiology solutions

Unless otherwise indicated, oocytes were recorded at room temperature in standard divalent ion-free bath solution (100 mM NaCl and

2 mM HEPES, pH 7.5), 2 mM $Ca^{2+}$-containing bath solution (100 mM NaCl, 2 mM $CaCl_2$, and 2 mM HEPES, pH 7.5), or 70 mM $Ca^{2+}$ bath solution (70 mM $CaCl_2$ and 2 mM HEPES, pH 7.5). For solutions used in the ion permeability assay, 100 mM NaCl in the standard divalent ion-free bath solution was replaced with 100 mM LiOH, CsCl, dimethylamine (DMA) hydrochloride, diethylamine (DEA) hydrochloride, tetraethylammonium (TEA) hydrochloride, or N-methyl-D-glucamine (NMDG) hydrochloride. The pH of $Li^+$ and $NMDG^+$ solutions was adjusted to 7.5 with HCl. The pH of the $Cs^+$, $DMA^+$, $DEA^+$, and $TEA^+$ solutions was adjusted to 7.5 with NMDG·HCl. Before application, 1 mM niflumic acid was added into these ion permeability assay solutions to inhibit unknown channel activity induced by some of these ions.

### Electrophysiology protocol and relative ion permeability ratio calculation

A standard TEVC protocol includes holding oocytes at 0 mV and measuring the current–voltage (I–V) relationships by applying an I–V protocol which runs 80-ms voltage steps from −100 to +100 mV (or indicated ranges) in 10-mV increments. In the experiments for determining ion permeability, oocytes were perfused with 100 mM $NMDG^+$ bath solution first, and the voltage step protocol was applied in order to extract the I–V curve and the reversal potential of $NMDG^+$. Oocytes were then washed with solutions containing 100 mM of the other ions tested. The I–V was monitored with a voltage ramp protocol (from −80 to +60 mV in 170 ms) until the reversal potential shift stopped (usually takes about 20s). Finally, another voltage step protocol was applied to extract the I–V curve and reversal potential for the tested ion. Relative permeability of the tested ions was calculated based on the difference between the reversal potentials of the tested ion and the reference ion ($Na^+$ or $NMDG^+$). The permeability ratios between tested ions $x^+$ and the reference ion ($Na^+$ or $NMDG^+$), $P_x/P_{Ref}$, were calculated using the following modified Goldman–Hodgkin–Katz (GHK) equation [59]:

$$P_X/P_{Ref} = e^{\Delta E_{rev}F/RT}$$

where $E_{rev}$ is the reversal potential, and $\Delta E_{rev}$ is the reversal potential difference between that of ion $x^+$ and the reference ion ($Na^+$ or $NMDG^+$). $\Delta E_{rev} = E_{rev,X} - E_{rev,ref}$. $F$ is Faraday's constant, $R$ is the universal gas constant, and $T$ is the absolute temperature.

### Co-immunoprecipitation (co-IP)

Twenty oocytes of each group were used for the co-IP experiment. After proper incubation, oocytes were washed with cold PBS solution three times and kept in a lysis solution containing 1x PBS, 1 mM EDTA, 10% glycerol, 1% n-dodecyl b-D-maltoside (DDM), and 1/50 (v/v) Protease Inhibitor Cocktail (Sigma). They were first homogenized by passing through a 25-G needle 10 times and then incubated by rotating at 4°C for 1 h. Lysates were centrifuged for 30 min at 10,000 *g*, and supernatants were collected. The supernatants were mixed with 20 μl anti-HA (Pierce) or anti-FLAG (Sigma) antibody-coated magnetic beads and rotated at 4°C for 2–5 h. The beads were collected and then washed with 400 ml wash solution (lysis solution plus 500 mM NaCl and 0.25% Triton X-100) three times by rotating at 4°C for 5 min. Proteins were then eluted

with 40 µl acid elution buffer (0.1 M glycine, pH 2.6) and neutralized with 1 M Tris pH 8.5, or eluted with 1.5× SDS loading buffer by incubating at 37°C for 30 min.

## Surface biotinylation

Proteins on *Xenopus* oocyte plasma membrane were detected with Pierce Cell Surface Protein Isolation Kit following a modified protocol described previously [67]. In brief, 2 or 4 days after cRNA injection, oocytes (20–30 oocytes per group) were washed with cold OR2 solution (82.4 mM NaCl, 2.5 mM KCl, 1 mM $MgCl_2$, 10 mM HEPES, pH 7.6). Oocytes were then incubated with 0.4 mg/ml sulfo-NHS-SS-biotin in ice-cold OR2 at 4°C for 30 min. The reaction was quenched, and oocytes were washed following the manufacturer's protocol. Oocytes were then homogenized and lysed. Lysates were mixed with NeutrAvidin Agarose at 4°C overnight. After beads were washed, proteins were eluted with 1.5× SDS sample loading buffer with 50 mM DTT at 37°C for 30 min. Eluted samples were analyzed by SDS–PAGE and Western blot. When required, the intensity of Western blot bands was measured with ImageJ (NIH).

## SDS–PAGE, Western blot, and antibodies

After TEVC recording, oocytes were collected, and protein expression was assessed by Western blot. Oocytes were lysed in the same way as in the co-IP experiment. Lysate samples were run on 4–12% Bolt Bis-Tris Plus Gels (Life Technologies) and transferred to PVDF membrane. Rabbit polyclonal anti-PC1 antibody recognizing aa 4,123–4,291 on the C-terminus of mouse PC1 [29], mouse monoclonal anti-PC1 N-terminus antibody 7e12 (Santa Cruz Biotechnology), mouse monoclonal anti-PC2 (YCE2, Santa Cruz Biotechnology), anti-HA (BioLegend), anti-FLAG (Sigma), or anti-β-actin antibody (GenScript) were used. Blot signals were visualized with the Molecular Imager ChemiDoc XRS+ Imaging System (Bio-Rad) or LI-COR Odyssey.

The protein expression of all mutations tested in this study has been confirmed by Western blot. All Western blots were repeated at least twice.

## $^{45}Ca$ uptake measurements

$^{45}Ca$ uptake measurements on oocytes were adapted from the previous description [55]. Briefly, $^{45}CaCl_2$ (PerkinElmer, Catalog No. NEZ013) at 30 µM and non-radioactive $CaCl_2$ at 1 mM were added to the uptake solution (100 mM NaCl, 2 mM KCl, 10 mM HEPES, pH 7.5). Ten oocytes per group were incubated with 0.3 ml uptake solution for 30 min. The uptake process was terminated by washing the oocytes with cold uptake solution without calcium five times. Individual oocytes were then lysed with 300 µl of 10% SDS. Four ml of scintillation cocktail (MP Biochemicals LLC. Catalog No. 88245305) was added to the lysate, and the radioactivity of each sample was measured with an LS6500 Multi-purpose Scintillation Counter (Beckman Coulter, Brea, CA). Data were analyzed by SigmaPlot 13.0.

## Structural graphics

The structural graphics were prepared with the software PyMOL (The PyMOL Molecular Graphics System).

## Statistical analysis

Recording data were analyzed with Excel or GraphPad Prism, and statistical significance was calculated with Student's *t*-test. Results of $P < 0.05$ were considered statistically significant (differences $P < 0.05$ is denoted by *, $P < 0.01$ by **, and $P < 0.001$ by ***). Results are presented as means ± SD.

## Animal use

Frogs care and experimental protocols were conducted under the approval of IACUC at St. John's University.

**Expanded View** for this article is available online.

## Acknowledgements

We thank members of the Yu laboratory for commenting on the manuscript, and Dr. Huangxue Xu in Qian Laboratory for technical support. This work was supported by National Institutes of Health Grants DK102092 (to Y.Y.), R01DK111611 (to F.Q.), and P30DKO90868 (to Baltimore PKD Center and to Y.Y. through pilot fund), The PKD Foundation Research Grant 230G18a (to Y.Y.), the Natural Sciences and Engineering Research Council of Canada DG RGPIN 401946 (to X.-Z.C.) and 05842 (to R.T.A.), and the Kidney Foundation of Canada BRG KFOC180027 (to X.-Z.C.).

## Author contributions

ZW and CN performed most of the biochemical and biophysical experiments. XL and RTA designed and performed the $^{45}Ca$ uptake experiment. YW and RW helped on making PC1 constructs. PK, HAC, BL, AC, EMK, and LI provided technical support. ZW, CN, XL, X-ZC, and YY analyzed data. YY, FQ, and X-ZC supervised the project. YY and ZW wrote the manuscript. All authors discussed the results and commented on the manuscript.

## Conflict of interest

The authors declare that they have no conflict of interest.

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
