## [Review Process File · EMBO Reports]

The ion channel function of polycystin-1 in the polycystin-1/polycystin-2 complex

Zhifei Wang, Courtney Ng, Xiong Liu, Yan Wang, Bin Li, Parul Kashyap, Haroon A. Chaudhry, Alexis Castro, Enessa M. Kalontar, Leah Ilyayev, Rebecca Walker, R. Todd Alexander, Feng Qian, Xing-Zhen Chen, Yong Yu

Review timeline:	Submission date:	23 April 2019
	Editorial Decision:	24 May 2019
	Revision received:	3 July 2019
	Editorial Decision:	30 July 2019
	Revision received:	30 July 2019
	Accepted:	1 August 2019

Transaction Report:

1st Editorial Decision

24 May 2019

Thank you for the submission of your research manuscript to our journal. We have now received the full set of referee reports that is copied below.

As you will see, the referees consider the presented findings of great interest and the evidence provided overall solid. However, the referees also have a number of suggestions how the study can be further strengthened, which should be addressed during the revision.

Given these constructive comments, we would thus like to invite you to revise your manuscript with the understanding that the referee concerns must be fully addressed and their suggestions taken on board. Please address all referee concerns in a complete point-by-point response. Acceptance of the manuscript will depend on a positive outcome of a second round of review. It is EMBO reports policy to allow a single round of revision only and acceptance or rejection of the manuscript will therefore depend on the completeness of your responses included in the next, final version of the manuscript.

Revised manuscripts should be submitted within three months of a request for revision; they will otherwise be treated as new submissions. Please contact us if a 3-months time frame is not sufficient for the revisions so that we can discuss the revisions further.

- 1) a .docx formatted version of the manuscript text (including legends for main figures, EV figures and tables). Please make sure that the changes are highlighted to be clearly visible.
- 2) individual production quality figure files as .eps, .tif, .jpg (one file per figure).

4) a complete author checklist, which you can download from our author guidelines (<<http://embor.embopress.org/authorguide>>). Please insert information in the checklist that is also reflected in the manuscript. The completed author checklist will also be part of the RPF.

5) Please note that all corresponding authors are required to supply an ORCID ID for their name upon submission of a revised manuscript (<<https://orcid.org/>>). Please find instructions on how to link your ORCID ID to your account in our manuscript tracking system in our Author guidelines (<<http://embor.embopress.org/authorguide>>).

6) We replaced Supplementary Information with Expanded View (EV) Figures and Tables that are collapsible/expandable online. A maximum of 5 EV Figures can be typeset. EV Figures should be cited as 'Figure EV1, Figure EV2' etc... in the text and their respective legends should be included in the main text after the legends of regular figures.

- For the figures that you do NOT wish to display as Expanded View figures, they should be bundled together with their legends in a single PDF file called *Appendix*, which should start with a short Table of Content. Appendix figures should be referred to in the main text as: "Appendix Figure S1, Appendix Figure S2" etc. See detailed instructions regarding expanded view here: <<http://embor.embopress.org/authorguide#expandedview>>.

7) We would also encourage you to include the source data for figure panels that show essential data. <optional: We would also encourage you to include the source data for the following figure panels:

- Figure xA
- Figure zB

Numerical data should be provided as individual .xls or .csv files (including a tab describing the data). For blots or microscopy, uncropped images should be submitted (using a zip archive if multiple images need to be supplied for one panel). Additional information on source data and instruction on how to label the files are available <<http://embor.embopress.org/authorguide#sourcedata>>.

8) Regarding data quantification, please ensure to specify the name of the statistical test used to generate error bars and P values, the number (n) of independent experiments underlying each data point (not replicate measures of one sample), and the test used to calculate p-values in each figure legend. Discussion of statistical methodology can be reported in the materials and methods section, but figure legends should contain a basic description of n, P and the test applied. Please note that error bars and statistical comparisons may only be applied to data obtained from at least three independent biological replicates.

I look forward to seeing a revised version of your manuscript when it is ready. Please let me know if you have questions or comments regarding the revision.

REFeree REPORTS

Referee #1:

The manuscript by Wang et al. describes the ion channel function of polycystin-1 (PC1) in the PC1/TRPP2 complex. The authors investigate the channel function of this disease-relevant protein complex in *Xenopus* oocytes. They show that the PC1/TRPP2 channel complex has biophysical properties that are distinct from homomeric TRPP2 channels. Using a combination of mutagenesis and electrophysiological experiments the authors present evidence that PC1 contributes to the PC1/TRPP2 channel pore. Furthermore, they show that the transmembrane domains of PC1 are sufficient for its channel activity and that GPS cleavage is not required for PC1's contribution to channel activity. The literature concerning the channel function of TRPP2 and PC1/TRPP2 is controversial, mostly because it has proven difficult to express functional polycystin proteins in heterologous expression systems. The authors have overcome this problem by introducing gain of function mutations in TRPP2 that show robust whole cell currents in *Xenopus* oocytes. The current study is based on previous work of the authors investigating the channel function of homomultimeric TRPP2 channels. Here they extend this work to investigate whether PC1 contributes to channel function in the PC1/TRPP2 complex. This is an important question since both proteins are thought to function in a complex, which is supported by genetic, biochemical, and structural data. There has been a debate about the permeation properties of the polycystin channel complex. While many reports have claimed that TRPP2 (with or without PC1) shows some Ca²⁺ permeability, a more recent study by the Clapham group suggests that TRPP2 has a very low Ca²⁺ permeability. The current study presents convincing evidence suggesting that PC1 increases the Ca²⁺ permeability of TRPP2 providing a potential explanation for conflicting results in the literature. Overall, these data represent an important advance in the PKD field. The manuscript is written clearly and the main conclusions are supported by the data. In my opinion, this is an excellent study which should be of very high interest to the PKD research community. Therefore, I recommend publication of this manuscript. I have a few questions and comments that the authors might want to address.

1. In Fig. 1B, the authors refer to the upper band of TRPP2 as being glycosylated (marked by a star). I would phrase this more carefully as "likely glycosylated" since no experiments are shown to prove that the upper band is glycosylated. Alternatively, the authors could perform experiments using deglycosylating enzymes to support this claim. I am not requesting these experiments as a requirement for publication though.

2. Fig. 1B: It appears as if non-glycosylated TRPP2 is present at the surface which is puzzling. Could the authors comment on this and briefly discuss it in the manuscript? Based on the electrophysiological experiments I am not questioning that the complex makes it to the plasma membrane. But seeing non-glycosylated TRPP2 at the surface raises questions about the surface labeling. Did the authors perform negative and positive controls (e.g. by expressing strictly intracellular proteins and known surface proteins) for the surface labeling that are not shown in the manuscript?

3. Fig. 3G: The authors show uptake of radio-labelled Ca²⁺ in oocytes expressing TRPP2 and PC1 WT. Since no currents can be recorded from WT channels it would be interesting to know whether Ca²⁺ uptake is increased in cells expressing the gain of function mutations.

4. Given the recent discussion about the permeation properties of TRPP2 and the reported high K⁺ permeability, I am wondering why the authors haven't tested the K⁺ permeability of homo- and heteromeric polycystin channels in their experimental setting.

5. Fig. 6: The authors generate pore mutations of PC1 and TRPP2 and state that these mutations change the ion selectivity of the PC1/TRPP2 complex. While it is true that there are changes in the permeation properties of the pore mutations, I am wondering whether these are true changes in the cation selectivity. The two large cations TEA and NMDG permeate very poorly through the channel, with NMDG being virtually impermeable. For all other cations the pore mutations result in reduced permeability. Couldn't this reduced permeability be explained by an overall decreased conductance of the channels with pore mutations. Fig. S7 suggests that conductance of the pore mutants is reduced indeed. A reduced conductance would not have a significant effect on cations that are not permeable (NMDG) or poorly permeable (TEA). If the authors agree that this might be an explanation, for the observed changes in changed ion selectivity, it should be discussed in the

manuscript. Alternatively, the authors should present data showing a true change in selectivity - e.g. pore mutations that turn the channel from a more K⁺-selective to a more Na⁺-selective channel.

6. Page 20, line 426: I would slightly tone down the following statement: "The fact that no TRPP2 was seen on surface when PKD1_L3040H was coexpressed also confirms that in our experimental condition, all TRPP2 are in the complex of PKD1 and no homomeric TRPP2 channel exists." I am not sure whether such a definitive statement can be made based on these data. I would like to suggest to replace "confirms" with "suggests" and "all" with "most" or "the majority of".

Minor comments:

- The authors use the abbreviation PKD1 for the Polycystin-1 protein. PKD1 usually refers to the gene name. I would rather use the common abbreviation PC1.
- Typos: Fig. 1B&C "Lysates" misspelled, line 885: TRP2_AA
- In line 33/34 and 563/564, the authors claim that their findings will aid the development of new therapeutic strategies for ADPKD. I strongly believe that the data presented here advance our understanding of channel function of the polycystin complex. I am not sure how this will aid the development of therapies. In my view such statements (which are found in many basic science papers today) are not helpful unless the authors provide insights how their finding may aid the development of therapeutic strategies.

Referee #2:

This manuscript describes an extensive set of experiments designed to whether the heterotetrameric polycystin-1/TRPP2 complex forms a functional ion channel. The authors present data suggesting that a channel activity for the PC1/2 complex can indeed be detected and they present some characterization of this channel's permeability properties. The authors show that, when it is expressed alone in *Xenopus* oocytes, a mutant form of TRPP2, TRPP2F604P, produces cation currents that are inhibited by co-expression with PC1. Inward current is inhibited by extracellular calcium. A TRPP2 double mutant (L677A/N681A; TRPP2AA) produces larger currents that are also inhibited by extracellular calcium. Removing extracellular cations except for 70 mM calcium eliminates inward current for TRPP2F604P, but TRPP2AA produces robust inward current under these conditions, which appears to be due to activation of calcium-activated chloride channel activity (CaCC). The authors conclude that TRPP2AA permits a small amount of calcium entry, which activates CaCC activity. Thus, CaCC activity amplifies the magnitude of a small TRPP2AA calcium entry that is otherwise too small to measure. Thus, the authors use CaCC activity as an indicator of calcium entry. Co-expression of TRPP2AA with PC1 produces a channel with very similar properties to TRPP2AA alone, except that the inward current is not blocked by extracellular calcium. Co-IP verifies incorporation of TRPP2AA into heteromeric channels with PC1. In 70 mM extracellular calcium PC1/TRPP2AA channels produced ~3X larger inward currents than did TRPP2AA alone. These inward currents were partially but not completely blocked by CaCC inhibitors. The authors conclude that PC1/TRPP2AA conducts calcium better than TRPP2AA alone. Inhibiting CaCC leaves more residual current with PC1/TRPP2AA than with TRPP2AA alone, and increasing extracellular calcium shifts the reversal potential to the right, indicating calcium conductance. Ca⁴⁵ uptake shows that calcium uptake is 6.5X higher for TRPPAA/PC1 than it is for TRPP2AA alone (which is barely above background). A PC1 CTF construct co-expressed with TRPP2AA produces currents very similar to WT PC1/TRPP2AA, and mutation of a putative PC1 pore lining residue (R4090W) blocks all current. The authors conclude that this mutagenesis proves that the calcium-mediated inward current is not due to activation of other endogenous channels. Similar (but smaller) currents are seen with a PC1 construct that includes only the extracellular S5-S6 loop and S6-S11. Both the CTF and S6-11 constructs appear to reach the cell surfaces of oocytes in association with TRPP2AA. Reversal potential measurements indicate that reversal potentials for organic cations are more positive with PC1/TRPP2AA than for TRPP2A alone, suggesting that the presence of PC1 alters pore properties and makes the channel more permeable to organic cations. Putative pore blockers (Ga, amiloride, ruthenium red) have different effects in TRPP2AA alone versus PC1/TRPP2AA, again suggesting differences in pore properties. Mutating putative pore forming residues in PC1 alters the selectivity and conductance of PC1/TRPP2AA, again suggesting that PC1 contributes to the pore. Finally, mutation of the GPS cleavage site produces mutation-dependent effects (L3040H, which has no rescue activity in mice *in vivo*, produces no current,

whereas T3041V, which produces partial in vivo rescue, supported channel activity and acquisition of ENDO H resistance).

As is evident from the preceding summary, the authors have amassed an extensive data set that supports their contention that the TRPP2AA/PC1 complex can function as an ion channel. This is an interesting result and has the potential to be an important contribution to a complex and controversial field. There are issues which, if addressed, would further strengthen the manuscript.

1) The manuscript would be much stronger if it provided additional insight into the single channel properties of the PC1/TRPP2AA channel. If obtainable, single channel recordings would be a tremendous addition to this story.

2) It would be very useful for the authors to provide within the Discussion section a summary, based upon the data that they have amassed, of the properties that the PC1/TRPP2 channel might be expected to exhibit under physiological conditions. When it is gated open by some as yet to be identified physiological stimulus, how much inward calcium current would the heterotetramer composed of wild type subunits be expected to carry? Is the magnitude of this current or of the associated calcium flux likely to be physiologically significant?

Minor: It would be helpful to indicate within Figure 3F that the experiments were performed in the presence of MONNA. Furthermore, the experiment depicted in Figure 4F should be repeated in the presence of MONNA.

Referee #3:

Mutations in PKD1 and PKD2 have been linked to kidney disease, ADPKD. Early studies have indicated that PKD1 forms cation channels in complex with PKD2 and a 1:3 stoichiometry has been suggested based on the structures of related PKD2L2 and more recently the complex cryo-EM high resolution structure of PKD1/PKD2 complex. However, exactly how the heteromeric channels function and the physiological significance of PKD1 contributions in the complex remain largely unexplored because of the difficulty in activating the PKD channels. In this report, the authors overcome this problem by taking advantage of newly identified PKD2 mutant, L677A/N681A, which is constitutively active when expressed in *Xenopus* oocytes. The interesting finding is that when co-expressed with either full-length or N-terminal truncated PKD1, the channels showed different Ca²⁺ permeability and selectivity to monovalent and organic cations from the homomeric PKD2 (mutant) channels, indicating that not only PKD1 participates in forming the channel pore but it also alters the ion selectivity as well as pore size of the channel. The contribution of PKD1 to the ion-conducting pore was further demonstrated by using mutants bearing amino acid substitutions at the putative pore-lining residues. Overall, the study provides solid evidence for the structural and functional contribution of PKD1 in the PKD1/PKD2 heteromeric channels and reveals a number of new biophysical features of the complex channels. The findings help advance our understanding of these important channels and will inspire new therapeutic strategies to treat ADPKD. I only have minor suggestions.

1) For PKD1-TLD, it appears that plasma membrane targeting of TRPP2-AA was impaired (Fig. 4J, anti-HA blot). Thus, the low current amplitude of the PKD1-TLD/TRPP2-AA may be explained by the deficit in surface expression of the functional channel. If this is true, then the result would indicate a role of S1-S5 of PKD1 in membrane targeting, rather than channel activity as stated in page 14, line 293.

2) Very high concentrations of Gd³⁺ (0.5 mM) and RuR (0.1 mM) were used to block TRPP2_AA and PKD1-CTF/TRPP2_AA channels. These concentrations tend to nonspecifically block many cation channels. Have you tried to use lower concentrations?

3) It is stated that mutation of PKD1 L4083 or V4085 to aspartic acid produced drastic effects on ion selectivity, but no information is given on the effect of the mutations on current amplitude. Did these mutants have similar current amplitude as the WT PKD1 when coexpressed with TRPP2-AA?

4) Figs. 1B 4I, 4J, 7D, western blots. Are the molecular weight markers supposed to be same between the two neighboring panels? The bands do not seem to align. Either the blots need realignment or markers should be labeled separately.

5) Grammar and word usage should be checked carefully. For example, page 10, line 213, "to further approve it" should be "to further prove it"; page 14, line 293, "more carefully comparison" should be "more careful comparison"; Fig. S2 legend, "slowly developement" should be "slow development".

6) Page 11, line 242-243, the sentence "the total amount of surface-expressed TRPP2 in

PKD1/TRPP2 sample is less than 1.5 folds more of that in TRPP2 alone" reads awkward. Consider revising.

7) Page 20, line 442, "The new GOF channel has..." should be "The new heteromeric GOF channel has..."

1st Revision - authors' response

3 July 2019

Response to Reviewer #1

Major points:

1. In Fig. 1B, the authors refer to the upper band of TRPP2 as being glycosylated (marked by a star). I would phrase this more carefully as "likely glycosylated" since no experiments are shown to prove that the upper band is glycosylated. Alternatively, the authors could perform experiments using de-glycosylating enzymes to support this claim. I am not requesting these experiments as a requirement for publication though.

Thank you for your comment. First of all, we have changed the names of PKD1 and TRPP2 in this manuscript to polycystin-1 (PC1) and polycystin-2 (PC2) respectively based on this reviewer's minor comment #1 below.

In a previous study from the coauthor Feng Qian's lab (cited in the manuscript), this 130 kDa band of PC2 has been identified as a higher glycosylated form of PC2 which is in complex with PC1. In that study, they employed de-glycosylating enzymes to treat the sample and found that the 130 kDa PC2 band is EndoH-resistant but can be de-glycosylated by PNGaseF [1]. We have therefore changed our sentence to "It has been previously found that the 130 kDa PC2 is a higher-glycosylated (EndoH resistant) form of PC2"

2. Fig. 1B: It appears as if non-glycosylated TRPP2 is present at the surface which is puzzling. Could the authors comment on this and briefly discuss it in the manuscript? Based on the electrophysiological experiments I am not questioning that the complex makes it to the plasma membrane. But seeing non-glycosylated TRPP2 at the surface raises questions about the surface labeling. Did the authors perform negative and positive controls (e.g. by expressing strictly intracellular proteins and known surface proteins) for the surface labeling that are not shown in the manuscript?

Based on what was found in Dr. Qian's paper mentioned above [1], the 120 kDa PC2 is also glycosylated, but its glycosylation is EndoH sensitive. Thus, the two forms of PC2 may take different routes in their trafficking to the plasma membrane. We have further edited the sentence to "It has been previously found that the 130 kDa PC2 is a higher-glycosylated (EndoH resistant) form of PC2 which stays in complex with PC1 in cilia of native tissues, while the 120 kDa PC2 only contains EndoH-sensitive glycosylation." In the description of Fig. 1C, we also clarified that the association between PC1 and both forms of PC2 were detected.

In our surface biotinylation experiments, and the major concern was to show that the PC2 proteins we detected are surface, instead of internal proteins. Thus, we have included the negative control actin (see the bottom images in Fig. 1B), which is more critical. Since actin was only detected in the oocyte lysate samples but not in the surface samples, this result proved that the PC2 protein in our surface sample is only from the plasma membrane.

Since we could detect PC2 on the surface, we feel that expressing another membrane protein as a positive control will not significantly improve our result. The best positive control would be an endogenous surface protein that is mainly located on the oocyte surface. However, we could not find a good target molecule, which can be detected specifically by an available antibody. We hope that the reviewer will, therefore, agree with us that the current negative control (actin) is sufficient for this experiment.

3. Fig. 3G: The authors show uptake of radio-labeled Ca^{2+} in oocytes expressing TRPP2 and PC1 WT. Since no currents can be recorded from WT channels, it would be interesting to know whether Ca^{2+} uptake is increased in cells expressing the gain of function mutations.

We thank the reviewer for this nice suggestion. We did not do this experiment initially since we were focusing on testing in our conclusion on Ca^{2+} permeability difference is true even in WT channels. We totally agree that the reviewer has a very good point about testing the GOF channels with the ^{45}Ca uptake experiment. If successful, it would strengthen our current conclusion. Thus, we have conducted two new ^{45}Ca uptake experiments with both WT and GOF channel.

First of all, the new results further proved what we have observed before. The ^{45}Ca uptake rate of oocytes expressing WT PC1/PC2 channel is consistently higher than oocytes expressing only WT PC2 channel (results shown below). However, due to some intrinsic difficulty of this experiment, we did not observe differences between WT PC1/PC2, PC2_AA, and PC1/PC2_AA channels (Results shown below). We believe this is due to the membrane potential of the oocytes and Ca^{2+} concentrations employed.

The membrane potential and concentration gradient (diffusion) are the two driving forces for cation influx. After 2-3 days of incubation of the oocytes after injection, the un-injected oocytes usually have their membrane potential at about -30 to -40 mV. The membrane potential of oocytes expressing WT PC2 or PC1/PC2 channels usually depolarized to above -25 mV. After expressing GOF PC2_AA, the membrane potential of the oocytes depolarized further to above -20 mV, and usually it shifts further to around -10 mV after expressing the GOF PC1/PC2_AA. The membrane potential shift most likely reflects an increase of membrane conductance (or leak) of the plasma membrane after expressing constitutively activated channels. When we did the electrophysiology recording experiment, we hyperpolarized the cell membrane to promote the cation influx. However, when we performed the ^{45}Ca uptake experiments, we cannot hyperpolarize the membrane potential to drive the cation influx. As can be seen from Fig. 3F, at a -20 to -10 mV membrane potential, the PC2_AA and the PC1/PC2_AA channels only have net outward current, which indicates that the cation influx at this voltages is very small [much smaller than cation (mainly K^+) efflux].

Also, when we did the electrophysiology recording, we were able to use a high concentration of Ca^{2+} (up to 70 mM) to increase the diffusion force to drive more cation influx (such as in Fig. 3D and F). However, we were not able to use a high concentration of Ca^{2+} when we performed the ^{45}Ca uptake experiments as these experiments are much longer and a high external calcium concentration for the duration of these experiments would damage the Oocyte. Thus, both driving forces are greatly reduced in the ^{45}Ca uptake experiments compared to that in our TEVC recording.

We did two rounds of new experiments, and the results are shown below. The difference between the two experiments is that we use 100 mM Na^+ in the bath in the first experiment but 100 mM NMDG⁺ in the second experiment. We made this change to reduce competition between Na^+ and Ca^{2+} in passing through the channel, in the hope of getting more Ca^{2+} influx. As you can see, even at the depolarized membrane potential, WT PC1/PC2 channel has significantly greater ^{45}Ca uptake than the WT PC2 group, as we reported in the current manuscript. The ^{45}Ca uptake rate of PC2_AA and PC1/PC2_AA channels are also significantly higher than that of the WT PC2 channel. However, although the last three groups (WT PC1/PC2, PC2_AA, and PC1/PC2_AA) all have significantly greater ^{45}Ca uptake than the WT PC2 channel, this experiment was not able to distinguish the difference between them. As we mentioned above, this is most likely due to the decrease of driving force for Ca^{2+} influx.

Thus, we concluded that the 45 Ca uptake experiment is not suitable for testing the Ca^{2+} influx difference between PC2_AA and PC1/PC2_AA channels. Since these experiments are not conclusive, we decided not to include the data in the manuscript. Also, since the 45 Ca uptake rate of WT PC1/PC2 channel is significantly higher than that of the WT PC2 channel in all our five experiments so far, we conclude that the difference between these two channels is big enough to be distinguished even under the current experimental conditions. Our conclusion on Ca^{2+} permeability difference between these two channels is still valid. We have added several sentences in the text (**lines 239-243**) to emphasize the potential influence of membrane potential on the 45 Ca uptake results.

Although we could not distinguish the Ca^{2+} permeability difference between PC2_AA and PC1/PC2_AA channels in the 45 Ca uptake experiments, we believe our recording data (Fig. 3B, D, and F) is sufficient to support our conclusion.

4. Given the recent discussion about the permeation properties of TRPP2 and the reported high K^{+} permeability, I am wondering why the authors haven't tested the K^{+} permeability of homo- and heteromeric polycystin channels in their experimental setting.

Due to the properties of the TEVC recording, we could not control intracellular ion concentration. If we assume that most of the cations inside of the oocytes are potassium (which is probably true [2]), the majority of the outward currents in our recordings are carried by K^{+} efflux. Since all channels studied in this manuscript have significant outward currents, they all, therefore, have significant K^{+} permeability. In the revised manuscript, we have added a sentence to point this out when we first describe the current of PC2_AA (**lines 162-164**). We thank the reviewer for his/her comment.

The reversal potentials of both PC2_AA and PC1/PC2_AA channels are negative when their currents were recorded in 100 mM Na^{+} solution (Fig. 3A), indicating that both channels are more permeable to K^{+} than Na^{+} . However, we are not able to calculate the exact permeability ratio of K^{+} to Na^{+} , due to the unknown concentration of intracellular K^{+} and other cations.

We can consider the intracellular K^{+} concentration is close to 100 mM [2], in this case, if we record channel current in a bath solution containing 100 mM KCl, the reversal potential, which is the membrane potential when net current is zero, should be close to 0 mV since there is no chemical gradient, and the current is only determined by membrane potential. This was indeed the case when we tested the KCl solution following the reviewer's suggestion for both PC2_AA and PC1/PC2 channels. The results are shown in the figure below. Compared to the reversal potentials in 100 mM Na^{+} , the reversal potentials in 100 mM K^{+} shifted to 0 mV for both channels. Thus, with the TEVC method, we are not able to detect the reversal potential difference for these channels in K^{+} solution.

If we could manipulate the ions on both the intracellular and extracellular sides, we would be able to measure the K^+ permeability. This can be done with the macro patch technique. However, it will take a long time to set up the method and optimize the conditions. Since we already know that both channels have significant K^+ permeability (and now mentioned this in the manuscript), and the ions tested in the current manuscript already serve the purpose for us to make the conclusion on the channels' permeability difference. We decided not to include K^+ permeability in the manuscript. Hopefully, the reviewer agrees with this.

5. Fig. 6: The authors generate pore mutations of PC1 and TRPP2 and state that these mutations change the ion selectivity of the PC1/TRPP2 complex. While it is true that there are changes in the permeation properties of the pore mutations, I am wondering whether these are true changes in the cation selectivity. The two large cations TEA and NMDG permeate very poorly through the channel, with NMDG being virtually impermeable. For all other cations, the pore mutations result in reduced permeability. Couldn't this reduced permeability be explained by an overall decreased conductance of the channels with pore mutations. Fig. S7 suggests that conductance of the pore mutants is reduced indeed. A reduced conductance would not have a significant effect on cations that are not permeable (NMDG) or poorly permeable (TEA). If the authors agree that this might be an explanation, for the observed changes in changed ion selectivity, it should be discussed in the manuscript. Alternatively, the authors should present data showing a true change in selectivity - e.g. pore mutations that turn the channel from a more K^+ -selective to a more Na^+ -selective channel.

We thank the reviewer for this insightful comment.

If we apply different ions on the intracellular and extracellular side of a channel, the apparent reversal potential of the I-V curve will reflect the permeability difference (selectivity) between the two ions. As we mentioned above, the majority of cation inside of oocytes is K^+ . Thus, the reversal potentials of the current we recorded in solutions with tested ions actually reflect the selectivity between the tested ions and K^+ . This is not affected by the overall conductance change of the channel. Thus, the shift in reversal potential reflects the change of the selectivity between the tested ion and K^+ . As an example, L4083D mutation in PC1 dramatically shifted the reversal potentials for almost all tested ions, indicating a change of selectivity between the tested ions and K^+ . The mutation did not reduce the outward current in Li^+ and DEA^+ solutions compared to the pseudo-WT channel (previous Fig. S8, current Fig. EV5). Thus, this mutation did not significantly reduce channel conductance, at least for K^+ .

If we know the exact concentration of intracellular K^+ of the recorded oocyte and assume no other ion has a significant concentration inside of the oocyte, with the Goldman-Hodgkin-Katz equation, we will be able to calculate the absolute permeability ratio of the tested ion and K^+ . However, since we do not know the exact intracellular concentration of K^+ , although we know the reversal potential shift reflects the selectivity change between the tested ions and K^+ , we are not able to directly calculate the selectivity difference between the two ions. However, in this case, we can still quantitatively calculate the permeability ratio of a tested ion and a reference ion by comparing both their permeability to K^+ . During the calculation, we are able to eliminate the unknown K^+ .

concentration and K^+ permeability from the equation and obtain the permeability ratio (selectivity) between the tested ion and the reference ion. This process can be simplified by doing the calculation with a modified Goldman-Hodgkin-Katz (GHK) equation [3], which we have provided in the method section. A similar method has been used previously [4,5].

We chose NMDG as the reference ion in the original manuscript since its permeability is almost not changed by pore mutations. Thus it serves as a good reference. However, we agree with the reviewer that calculating the selectivity to a small reference ion will be valuable to further demonstrate the selectivity changes caused by the pore mutations. Thus, we calculated the permeability ratio of all tested ions relative to Na (P_x/P_{Na}), and added the new data to Figs. 5 and 6. We also moved the P_x/P_{NMDG} data from Fig S8 (currently Fig EV5) to Fig. 6. Description of these results has also been added to the main text (**lines 319-331; 371-376; 386-395**), and the related figure legends are also changed. We found, in most cases, pore mutations also cause changes in P_x/P_{Na} . Most interestingly, we found some mutations lead to an increase in this ratio of some ions but at the same time decrease of this ratio of the other ions. These result clearly show ion selectivity changes caused by these mutations, which is independent of overall channel conductance change.

In the revised manuscript, we have also added a couple of sentences to explain the meaning of reversal potential shift (**lines 305-308**). We thank the reviewer for this question, which, we believe, has led to an enhanced presentation of our results in the revised manuscript.

6. Page 20, line 426: I would slightly tone down the following statement: "The fact that no TRPP2 was seen on the surface when PKD1_L3040H was coexpressed also confirms that in our experimental condition, all TRPP2 are in the complex of PKD1 and no homomeric TRPP2 channel exists." I am not sure whether such a definitive statement can be made based on these data. I would like to suggest to replace "confirms" with "suggests" and "all" with "most" or "the majority of".

We agree with the reviewer and have made the changes in the text.

Minor comments:

1. The authors use the abbreviation PKD1 for the Polycystin-1 protein. PKD1 usually refers to the gene name. I would rather use the common abbreviation PC1.

We agree with the reviewer that PC1 is a better name for this protein in this manuscript and have made the changes in our text. Also, we feel that the use of "polycystin-2 (PC2)" instead of "TRPP2" will help to further clarify the relationship between these two proteins and help to avoid confusion (some people prefer to use TRPP2 to name polycystin-L). Thus, we also changed "TRPP2" to "PC2" in our manuscript.

2. Typos: Fig. 1B&C "Lysates" misspelled, line 885: TRP2_AA

We have made the changes.

3. In line 33/34 and 563/564, the authors claim that their findings will aid the development of new therapeutic strategies for ADPKD. I strongly believe that the data presented here advance our understanding of the channel function of the polycystin complex. I am not sure how this will aid the development of therapies. In my view, such statements (which are found in many basic science papers today) are not helpful unless the authors provide insights how their finding may aid the development of therapeutic strategies.

We agree with the reviewer and have deleted the claim.

Response to Reviewer #2

Major points:

1) The manuscript would be much stronger if it provided additional insight into the single channel properties of the PC1/TRPP2AA channel. If obtainable, single channel recordings would be a tremendous addition to this story.

We thank the reviewer for this suggestion and totally agree that single channel recordings would provide more insight into the channel properties of PC2_AA and PC1/PC2 channels (We decided to use the names “polycystin-1 (PC1)” and “polycystin-2 (PC2)” to replace “PKD1” and “TRPP2” in the first manuscript. Please see our response to the minor comment #1 of the first reviewer). Unfortunately, we had met difficulty previously in trying to do single channel recording of our GOF PC2_F604P channel through collaboration with Dr. Lawrence Palmer at Cornell University. In our recordings, several channels with different properties were found in the oocyte plasma membrane. Unfortunately, we were not confident to say that a certain kind of single-channel current we recorded is definitely conducted by the GOF PC2 channel. We believe several possible reasons may have led to the difficulty we met, such as the plasma membrane expression of this channel is low which made it hard to be patched, or the current of GOF PC2 is too similar to another endogenous channel in oocyte so we could not distinguish them, or the single channel conductance is too low.

At the same time, we believe that the absence of single channel data does not affect our main discovery and conclusions. Hopefully, the reviewers will understand and agree with us.

2) It would be very useful for the authors to provide within the Discussion section a summary, based upon the data that they have amassed, of the properties that the PC1/TRPP2 channel might be expected to exhibit under physiological conditions. When it is gated open by some as yet to be identified physiological stimulus, how much inward calcium current would the heterotetramer composed of wild type subunits be expected to carry? Is the magnitude of this current or of the associated calcium flux likely to be physiologically significant?

Thanks to the reviewer for this suggestion. We have added some discussion about the potential physiological significance of the Ca^{2+} influx through PC1/PC2 channel, assuming that they are located on cilia. However, just based on our current data, it is impossible to calculate how much Ca^{2+} influx would occur after the WT channel is gated by a stimulus. First of all, we do not know whether gated WT channels have similar conductance to the GOF channel; second, we do not know the numbers of channels on each cilium; third, without the signal channel recording data, we are not able to measure the single channel Ca^{2+} conductance of the channel.

Minor point:

1. It would be helpful to indicate within Figure 3F that the experiments were performed in the presence of MONNA. Furthermore, the experiment depicted in Figure 4F should be repeated in the presence of MONNA.

We thank the reviewer for these comments. The presence of MONNA has been indicated in the legend of Fig. 3F. To make it clearer, we have also indicated this information in the figure directly in the revised manuscript. Also, we assume that the reviewer wants us to repeat Fig. 4H, instead of 4F, in the presence of MONNA, since 4H was recorded in 70 mM Ca^{2+} , but not 4F. In the revised manuscript, we have added the currents showing the blocking of MONNA on both PC1-CTF/PC2_AA and PC1-LTD/PC2_AA currents in 70 mM Ca^{2+} in the new Appendix Fig 3.

Response to Reviewer #3

1) For PKD1-TLD, it appears that plasma membrane targeting of TRPP2-AA was impaired (Fig. 4J, anti-HA blot). Thus, the low current amplitude of the PKD1-TLD/TRPP2-AA may be explained by the deficit in surface expression of the functional channel. If this is true, then the result would indicate a role of S1-S5 of PKD1 in membrane targeting, rather than channel activity as stated in page 14, line 293.

Thanks to the reviewer for this valuable comment. To address this point, we have repeated the surface biotinylation experiment three more times. We keep getting relatively lower PC2 expression when it is coexpressed with PC1-LTD. Thus, it appears as if the PC1-LTD/PC2 complex has less

trafficking to the plasma membrane compared to the other two longer constructs. We have replaced the Fig. 4J with new western blot images and clarified this point in the revised text (**lines 284-289**). We believe this does not affect our claim that the PC1-LTD/PC2 complex is a functional channel when it is expressed in the plasma membrane, and the LTD domain is sufficient for channel function of PC1.

2) Very high concentrations of Gd³⁺ (0.5 mM) and RuR (0.1 mM) were used to block TRPP2_AA and PKD1-CTF/TRPP2_AA channels. These concentrations tend to nonspecifically block many cation channels. Have you tried to use lower concentrations?

In the original experiments, we were just trying to figure out the different effects of the blocker on these two channels and wanted to make sure we completely inhibited them. However, we agree with the reviewer that having the data with lower concentrations of these blockers would provide more information. We have therefore repeated the experiments with the 5-10 times reduced concentrations of the blockers (0.1 mM Gd³⁺, 0.01 mM RuR, and 1 mM Amiloride). The new data with both high and low concentrations of blockers were used to replace the old data, and are currently presented in Fig. EV4. Discussion on these results is added in Results (**Lines 338-346**).

3) It is stated that mutation of PKD1 L4083 or V4085 to aspartic acid produced drastic effects on ion selectivity, but no information is given on the effect of the mutations on current amplitude. Did these mutants have similar current amplitude as the WT PKD1 when coexpressed with TRPP2-AA?

The representative currents of both mutations and all other mutations had been included in Fig. S7 of the original submitted manuscript (current Fig. EV5). In most cases, L4083D did not change the current amplitude too much, while V4085D caused a reduction of the current size. We need to keep in mind that current size can be affected by the expression level of the particular protein, which is variable in different oocytes. However, since the reversal potential is the most critical information for us to calculate the permeability ratio, which is not affected by overall current size (please see our answer to comment #5 of reviewer 1), we did not consider current amplitude as long as the mutants provided significant current.

4) Figs. 1B 4I, 4J, 7D, western blots. Are the molecular weight markers supposed to be same between the two neighboring panels? The bands do not seem to align. Either the blots need realignment or makers should be labeled separately.

Yes, the markers are the same between the two neighboring panels. In most of our experiments, the proteins ran higher in the co-IP or surface biotinylation elution samples. We mentioned this in the legend of Fig. 1B in the original manuscript. It is most likely due to a higher concentration of salt used for elution. To make sure readers will notice it for sure, we have moved this sentence to the main text (**lines 132-134**).

5) Grammar and word usage should be checked carefully. For example, page 10, line 213, "to further approve it" should be "to further prove it"; page 14, line 293, "more carefully comparison" should be "more careful comparison"; Fig. S2 legend, "slowly development" should be "slow development".

We appreciate the careful review. All these mistakes have been corrected, and we also carefully checked the whole manuscript.

6) Page 11, line 242-243, the sentence "the total amount of surface-expressed TRPP2 in PKD1/TRPP2 sample is less than 1.5 folds more of that in TRPP2 alone" reads awkward. Consider revising.

Thank the reviewer for catching this. We have changed the sentence to "Under the designed experimental conditions, the total amount of surface-expressed PC2 in the PC1/PC2 sample is about 1.5 fold of that in the PC2 alone sample..."

7) Page 20, line 442, "The new GOF channel has..." should be "The new heteromeric GOF channel has..."

We have changed it.

References:

1. Kim H, Xu H, Yao Q, Li W, Huang Q, Outeda P, Cebotaru V, Chiaravalli M, Boletta A, Piontek K, *et al.* (2014) Ciliary membrane proteins traffic through the Golgi via a Rabep1/GGA1/Arl3-dependent mechanism. *Nature communications* **5**: 5482
2. Costa PF, Emilio MG, Fernandes PL, Ferreira HG, Ferreira KG (1989) Determination of ionic permeability coefficients of the plasma membrane of *Xenopus laevis* oocytes under voltage clamp. *The Journal of physiology* **413**: 199-211
3. Hille B (2001) *Ion Channels of Excitable Membranes* pp Sinauer Associates
4. Arif Pavel M, Lv C, Ng C, Yang L, Kashyap P, Lam C, Valentino V, Fung HY, Campbell T, Moller SG, *et al.* (2016) Function and regulation of TRPP2 ion channel revealed by a gain-of-function mutant. *Proceedings of the National Academy of Sciences of the United States of America* **113**: E2363-2372
5. Yu Y, Ulbrich MH, Li MH, Dobbins S, Zhang WK, Tong L, Isacoff EY, Yang J (2012) Molecular mechanism of the assembly of an acid-sensing receptor ion channel complex. *Nature communications* **3**: 1252

2nd Editorial Decision

30 July 2019

Thank you for the submission of your revised manuscript to EMBO reports. We have now received the full set of referee reports that is copied below.

As you will see, all referees are very positive about the study and support publication in EMBO reports.

Browsing through the manuscript myself, I noticed a few editorial things that we need before we can proceed with the official acceptance of your study.

- 1) Parul Kashyap is not listed in the Author Contributions paragraph.
- 2) Please enter the funder information also in our online submission portal (National Institutes of Health Grant P30DKO90868; Baltimore PKD Center Pilot & Feasibility Grant)
- 3) Figure 4E: the source data does not match the blots in the figure. The bands labeled with anti-PC1 in the source data are shown as PC2 in the figure and what is PC2 in the source data appears as PC1 in the figure. Please double-check the blots and their labels.
- 4) Figure 4J/S4: in the figure you indicate that PC2 was detected using an anti-HA antibody, while the source data file indicates that anti-YCE2 was used. Please double-check the labeling.
- 5) Our data editors from Wiley have already inspected the Figure legends for completeness and accuracy. I have also taken the liberty to make some changes to the Abstract. Please see all suggested changes in the attached Word file.
- 6) Finally, EMBO Press is pleased to support the "minimum reporting standards in the life sciences" initiative (<https://osf.io/preprints/metaarxiv/9sm4x/>). This effort brings together a number of leading journals and reproducibility experts to develop minimum expectations for reporting information about Materials (including data and code), Design, Analysis and Reporting (MDAR) in published papers. We believe broad alignment on these standards will be to the benefit of authors, reviewers, journals and the wider research community and will help drive better practise in publishing reproducible research.

We are therefore participating in a community pilot involving a small number of life science

journals to test the MDAR checklist. The checklist is intended to help authors, reviewers and editors adopt and implement the minimum reporting framework.

We very much hope that you will be willing to participate in this trial; the MDAR reporting checklist and an MDAR elaboration document providing context for the standards is attached to this message. If you agree to participate, please complete the MDAR reporting checklist and return it to us within 7 days. We would also be very grateful if you could complete this author survey <https://forms.gle/FRx7hpKS8g1QMNPR9>.

Please note that your completed checklist and survey will be shared with the minimum reporting standards working group. However, the working group will not be provided with access to the manuscript or any other confidential information including author identities, manuscript titles or abstracts. Feedback from this process will be used to consider next steps, which might include revisions to the content of the checklist. Data and materials from this initial trial will be publicly shared in September 2019. Data will only be provided in aggregate form and will not be parsed by individual article or by journal, so as to respect the confidentiality of responses.

Please treat the checklist and elaboration as confidential as public release is planned for September 2019.

If you decide against participating, we would be grateful for any feedback you may have.

REFEREE REPORTS

Referee #1:

The authors have addressed all my concerns and I recommend publication of the manuscript.

With respect to my comment 3 concerning the ^{45}Ca uptake experiments, I appreciate the effort of the authors to perform additional experiments. I would like to point out though, that I don't think that the explanation provided by the authors to explain a lack of differences between PC1/PC2 wt, PC2AA and PC1/PC2AA in the ^{45}Ca uptake experiments is very likely. The chemical driving force for Ca^{2+} under physiological conditions is rather high (reversal potential +120mV). The small observed differences in membrane potential of oocytes expressing the respective proteins are unlikely to explain the lack of increased ^{45}Ca uptake if there is a considerable increase in Ca^{2+} permeability/conductance of PC1/PC2AA as suggested by the electrophysiological experiments. Importantly, the experiments shown in the authors' response to comment 3 using NMDG instead of Na^{+} (2nd) further question this explanation since the oocytes have a more hyperpolarized membrane potential in the presence of NMDG which should increase the driving force for Ca^{2+} entry. However, I agree with the authors that their conclusion on the differences for the wt channels as shown in the manuscript is still valid.

Overall, I think that this is an excellent manuscript that will be of great interest to the scientific community.

Referee #2:

The authors have adequately addressed the comments from the previous review and the manuscript has been strengthened as a consequence of this effort.

Referee #3:

The authors have satisfactorily addressed all my concerns and the paper should be acceptable for

publication.

2nd Revision - authors' response

30 July 2019

The authors performed all minor editorial changes.

Corresponding Author Name: Yong Yu

Manuscript Number: EMBOR-2019-48336